# Circumventing Concept Erasure Methods For Text-to-Image Generative Models

**Minh Pham, Kelly O. Marshall, Niv Cohen, Govind Mittal & Chinmay Hegde**
New York University
{mp5847, km3888, nc3468, mittal, chinmay.h}@nyu.edu

## Abstract

Text-to-image generative models can produce photo-realistic images for an extremely broad range of concepts, and their usage has proliferated widely among the general public. Yet, these models have numerous drawbacks, including their potential to generate images featuring sexually explicit content, mirror artistic styles without permission, or even hallucinate (or deepfake) the likenesses of celebrities. Consequently, various methods have been proposed in order to "erase" sensitive concepts from text-to-image models. In this work, we examine seven recently proposed concept erasure methods, and show that targeted concepts are not fully excised from any of these methods. Specifically, we devise an algorithm to learn special input word embeddings that can retrieve "erased" concepts from the sanitized models with no alterations to their weights. Our results highlight the brittleness of post hoc concept erasure methods, and call into question their use in the algorithmic toolkit for AI safety.

## 1 Introduction

**Motivation.** Text-to-image models (Chang et al., 2023; Gafni et al., 2022; Ramesh et al., 2021; 2022; Saharia et al., 2022; Yu et al., 2022; Rombach et al., 2022; Xu et al., 2022) have garnered significant attention due to their exceptional ability to synthesize high-quality images based on text prompts. Such models, most prominently Stable Diffusion (SD) ( Rombach et al. (2022)) and DALL-E 2 ( Ramesh et al. (2022)), have been adopted in a variety of commercial products spanning application realms ranging from digital advertising to graphics to game design. In particular, the open-sourcing of Stable Diffusion has democratized the landscape of image generation technology. This shift underlines the growing practical relevance of these models in diverse real-world applications. However, despite their burgeoning popularity, these models come with serious caveats: they have been shown to produce copyrighted, unauthorized, biased, and potentially unsafe content (Mishkin et al., 2022; Rando et al., 2022).

What is the best way to ensure that text-to-image models do not produce sensitive or unsafe concepts? Dataset pre-filtering (AI, 2022) may present the most obvious answer. However, existing filtering procedures are imperfect and may exhibit a large proportion of false negatives. See the extensive studies reported in Birhane et al. (2021) on how LAION-400M, a common dataset used in training text-image models, contains numerous offensive image samples which persist after applying standard NSFW filters.

Even if perfect data pre-filtering were possible, substantial resources would be required to retrain large models from scratch in response to issues unearthed post-training. As a result, several *post hoc* concept-erasure methods have emerged of late. Some advocate inference guidance (Schramowski et al., 2023; AUTOMATIC1111, 2022). Others require fine-tuning the weights on an auxiliary subset of training data ( Gandikota et al. (2023a); Heng & Soh (2023); Zhang et al. (2023)). These may be categorized as more practical alternatives to full model-retraining with a stripped-down version of the original training data. Many of these methods are accompanied by public releases of the weights of the "sanitized" models. Such concept erasure methods are purported "to permanently remove [targeted concepts] from the weights"; moreover, they are presented as "not easy to circumvent since [the method] modifies weights" (Gandikota et al., 2023a). An array of results on several test instances

across use cases (object removal, artistic style forgetting, avoidance of NSFW content, avoiding likeness of specific people) seem to support the efficacy of these methods.

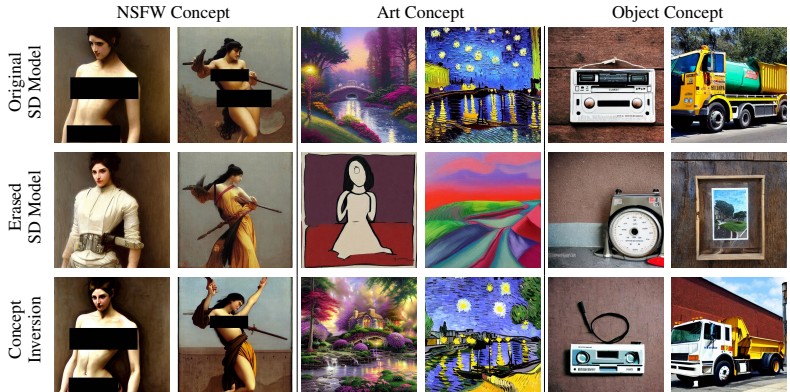

Figure 1: **Concept erasure methods fail to excise concepts from text-to-image models.** This figure shows results from ESD (Gandikota et al., 2023a), which is a variant of Stable Diffusion trained to avoid generating NSFW content, specific artist styles, and specific objects, like trucks ($2^{nd}$ row). We circumvent this method by generating the "erased" concepts ($3^{rd}$ row) by designing special prompts.

**Our contributions.** Our main contribution in this paper is to show that:

*Post hoc concept erasure in generative models provides a false sense of security.*

We investigate seven recently announced concept-erasure methods for text-to-image generative models: (i) Erased Stable Diffusion (Gandikota et al., 2023a), (ii) Selective Amnesia (Heng & Soh, 2023), (iii) Forget-me-not (Zhang et al., 2023), (iv) Ablating Concepts (Kumari et al., 2023), (v) Unified Concept Editing (Gandikota et al., 2023b), (vi) Negative Prompt (AUTOMATIC1111, 2022), and (vii) Safe Latent Diffusion (Schramowski et al., 2023). All of these were either published or appeared online in the first 9 months of 2023.

Somewhat surprisingly, we show that all seven techniques can be circumvented. In all cases, the very same "concept-erased" models — with zero extra training or fine-tuning — may produce the erased concept with a suitably constructed (soft) prompt. Therefore, the seemingly-safe model may still be used to produce sensitive or offensive content. Overall, our results indicate that there may be a fundamental brittleness to post hoc erasure methods, and entirely new approaches for building (and evaluating) safe generative models may be necessary. See Figure 1 for examples.

**Techniques.** Our approach stems from the hypothesis that existing concept erasure methods may be, in reality, performing some form of *input filtering*. More specifically, in these methods, the modified generative models produced by these methods are evaluated on a limited subset of text inputs: the original offending/sensitive text, and related prompts. However, this leaves the model vulnerable to more sophisticated text prompts. In particular, we design individual *Concept Inversion* (CI) "attack" techniques to discover special word embeddings that can recover erased concepts when fed to the modified model. Through the application of CI, we provide evidence that these unique word embeddings outmaneuver concept erasure methods across various use cases such as facial likeness, artistic style, object-types, and NSFW concepts. Therefore, it is not the case that these concepts have been permanently removed from the model; these still persist, albeit remapped to new embeddings.

**Implications.** Our extensive experiments below highlight two key points:

1. Our results call into question the premise that existing erasure methods (fully) excise concepts from the model. Our results show that this premise is not correct and that the results in these previous works on concept erasure should be scrutinized carefully.

2. We call for stronger evaluation methodologies for concept erasure methods. Measuring the degree of concept erasure in text-to-image models is tricky, since there are potentially a vast number of

prompts that a motivated (and moderately well-equipped) attacker can use as inputs. As a first step to mitigate this issue, we recommend evaluating models in terms of our CI attacks during evaluation, and not merely limited to evaluating over mild variations of the original text prompts.

Overall, our findings shine a spotlight on the considerable challenges in sanitizing already trained generative AI models (such as Stable Diffusion) and making them safe for wide public use.

## 2 BACKGROUND

**Denoising Diffusion Models.** Diffusion models belong to a category of generative models that sample from a distribution via an iterative Markov-based denoising process (Sohl-Dickstein et al., 2015; Ho et al., 2020). The process begins with a sampled Gaussian noise vector, denoted as $x_T$, and undergoes a series of $T$ denoising steps to ultimately restore the final data, referred to as $x_0$. In practical applications, the diffusion model is trained to predict the noise $\epsilon_t$ at each timestep, $t$, utilized to generate the progressively denoised image, $x_t$. Latent diffusion models (LDM) (Rombach et al., 2022) offer improved efficiency by operating in a lower dimensional space learned by an autoencoder. The first component of LDM consists of an encoder $\mathcal{E}$ and a decoder $\mathcal{D}$ that have been pre-trained on a large collection of images. During the training of LDM, for an image $x$, the encoder learns to map $x$ into a spatial latent code $z = \mathcal{E}(x)$. The decoder maps such latent codes back to the original images such that $\mathcal{D}(\mathcal{E}(x)) \approx x$. The second component is a diffusion model trained to produce codes within the learned latent space. Given a conditional input $c$, the LDM is trained using the following objective function:

$$\mathcal{L} = \mathbb{E}_{z \sim \mathcal{E}(x), t, c, \epsilon \sim \mathcal{N}(0,1)} \left[ \|\epsilon - \epsilon_\theta(z_t, c, t)\|_2^2 \right], \tag{1}$$

Here $z_t$ is the latent code for time $t$, and $\epsilon_\theta$ is the denoising network. At inference time, a random noise tensor is sampled and gradually denoised to produce a latent $z_0$, which is then transformed into an image through the pre-trained decoder such that $x' = \mathcal{D}(z_0)$.

Ho & Salimans (2022) propose a classifier-free guidance technique is used during inference and requires that the model be jointly trained on both conditional and unconditional denoising. The unconditional and conditional scores are used to create the final latent $z_0$. There, we start with $z_T \sim \mathcal{N}(0, 1)$ which is transformed to obtain $\tilde{\epsilon}_\theta(z_t, c, t) = \epsilon_\theta(z_t, t) + \alpha(\epsilon_\theta(z_t, c, t) - \epsilon_\theta(z_t, t))$, to get $z_{T-1}$. This process is repeated sequentially until $z_0$ is produced.

**Machine Unlearning.** The conventional goal in machine learning is to foster generalization while minimizing reliance on direct memorization. However, contemporary large-scale models possess the capacity for explicit memorization, whether employed intentionally or as an inadvertent byproduct (Carlini et al., 2023; Somepalli et al., 2023b;a). The possibility of such memorization has led to the development of many works in machine unlearning (Golatkar et al., 2020a;b), the core aim of which is to refine the model to behave as though a specific set of training data was never presented.

**Mitigating Undesirable Image Generation.** Numerous methods have been proposed to discourage the creation of undesirable images by generative models. One initial approach is to exclude certain subsets of the training data. However, this solution can necessitate the retraining of large-scale models from scratch, which can be prohibitive. An alternative put forward by Schramowski et al. (2023); AUTOMATIC1111 (2022) involves manipulating the inference process in a way that steers the final output away from the target concepts. Yet another approach employs classifiers to alter the output (Rando et al., 2022; AI, 2022; Bedapudi, 2022). Since inference guiding methods can be evaded with sufficient access to model parameters (SmithMano, 2022), subsequent works (Gandikota et al., 2023a; Heng & Soh, 2023; Zhang et al., 2023; Kumari et al., 2023; Gandikota et al., 2023b) suggest fine-tuning Stable Diffusion models. Qu et al. (2023) study the capability of generating unsafe images and hateful memes of various text-to-image models.

**Diffusion-based Inversion.** Image manipulation with generative networks often requires *inversion* (Zhu et al., 2016; Xia et al., 2021), the process of finding a latent representation that corresponds to a given image. For diffusion models, (Dhariwal & Nichol, 2021) demonstrate that the DDIM (Song et al., 2021) sampling process can be inverted in a closed-form manner, extracting a latent noise map that will produce a given real image. More recent works (Ruiz et al., 2023; Gal et al., 2023; Shi et al., 2023; Han et al., 2023) try to invert a user-provided concept to a new pseudo-word in the model's

vocabulary. The most relevant approach for our work is Textual Inversion (Gal et al., 2023)) which learns to capture the user-provided concept by representing it through new "words" in the embedding space of a frozen text-to-image model without changing the model weights. In particular, the authors designate a placeholder string, $c_*$, to represent the new concept the user wishes to learn. They replace the vector associated with the tokenized string with a learned embedding $v_*$, in essence "injecting" the concept into the model vocabulary. The technique is referred to as Textual Inversion and consists of finding an approximate solution to the following optimization problem:

$$v_* = \arg\min_v \mathbb{E}_{z \sim \mathcal{E}(x), c_*, \epsilon \sim \mathcal{N}(0,1), t} \left[ \| \epsilon - \epsilon_\theta(z_t, c_*, t) \|_2^2 \right].$$

## 3 PRELIMINARIES

**Basic setup and threat model.** For the remainder of the paper, we will leverage inversion techniques to design an "attack" on concept-erased models. We assume the adversary has: (1) access to the weights and components of the erased model, (2) knowledge of the erasure method, (3) access to example images with the targeted concept (say via an image search engine), and (4) moderately significant computational power.

A trivial approach to "un-erase" an erased concept would be via fine-tuning a sanitized model on sufficiently many example images. Therefore, we also assume that: (5) the adversary cannot modify the weights of the erased model.

To show that our CI attack is a reliable tool for establishing the existence of concepts in a model, we conduct two experiments to investigate whether Textual Inversion (TI) by itself can generate a concept that the model has not captured during training. If TI can hallucinate totally novel concepts, then even data filtering before training might not be able to avoid producing harmful/copyrighted content. In the first experiment, we compare TI performance on concepts that are better represented in the training data of Stable Diffusion 1.4, versus those that are likely not present. In the second experiment, we conducted a more controlled study by training two diffusion models on MNIST (LeCun et al., 2010) from scratch. We include all the training classes in the first run and exclude one class in the second run. In both experiments, we find that Textual Inversion works significantly worse when the concept is not well represented in the training set of the generative model. See Figure 15 in the Appendix.

## 4 CIRCUMVENTING CONCEPT ERASURE

### 4.1 EXPERIMENTAL SETUP

In this section, we examine seven (7) different concept erasure methods. To the best of our knowledge, this list constitutes all the concept erasure methods for Stable Diffusion models published up to September 19, 2023. We design CI procedures tailored to each erasure method that search the space of word embeddings to recreate the (purportedly) erased visual concepts. Importantly, our approach relies solely on the existing components of the post-erasure diffusion models. For these experiments, wherever possible we use the pre-trained models released by the authors unless explicitly stated otherwise; for concepts where erased models were not publicly available, we used public code released as-is by the authors to reproduce their erasure procedure. In each subsection, we start by describing the approach, then show how to attack their approach using Concept Inversion. We interleave these with results, and reflect on their implications. Finally, we show evidence that current concept erasure methods are likely performing input filtering, and demonstrate transferability of the learned word embeddings.

Due to page limit constraints, we will focus on results for two fine-tuning-based methods (ESD, UCE) as well two inference-guiding-based methods (NP, SLD). We refer readers to the full set of results for all seven models in Appendix C. [1]

### 4.2 EVALUATION PROTOCOL

For each concept erasure method that we will be discussing below, we initially deploy it to erase 4 concept categories including art style, object, ID, and NSFW content. We use Stable Diffusion

---

[1] Our code is available for reproducibility purposes at https://nyu-dice-lab.github.io/CCE/

1.4 (SD 1.4) for all our experiments. We assume that the adversary can access a small number of examples of the targeted concept from Google Images; see Appendix for details.

**Art style:** We select 6 styles from modern artists and artistic topics that have been reported to have been captured by SD 1.4: the movie series "Ajin: Demi Human", Thomas Kinkade, Tyler Edlin, Van Gogh, Kelly McKernan, and Tyler Edlin. We generate images from the erased models using the prompt "A painting in the style of [*artist name*]". After performing CI, we generate images by replacing [*artist name*] with $c_*$ - the special placeholder string associated with the learned word embedding. In addition to qualitative results, we follow Gandikota et al. (2023a) and conduct a human study to measure the effectiveness of our CI methods. In particular, for each artist, we collect 10 images of art created by those artists from Google Images. We then generate 10 images from the erased model using the standard concept name, and 10 images using CI per style and per concept erasure method. Participants were shown 5 real reference images from the same artist and another image of the same style (either real, from the erased model or from CI). They were then asked to estimate, on a five-point Likert scale, their confidence level that the experimental image has the same style as the reference images. Our study consists of 50 participants, with 96 responses per participant.

**Objects:** Following Gandikota et al. (2023a), we investigate the Imagenette (Howard, 2019) dataset which comprises ten easily identifiable classes (cassette player, chain saw, church, etc.) We evaluate CI methods by examining the top-1 predictions of a ResNet-50 Imagenet classifier on 500 generated images. We generate images from the erased models using the prompt "A photo of a [*object name*]". For CI, we generate images by replacing [*object name*] with the special string $c_*$.

**ID:** Following Heng & Soh (2023), we select "Brad Pitt" and "Angelina Jolie" as identity concepts. We then utilize the GIPHY celebrity detector (Giphy, 2020) for Concept Inversion evaluation. We generate 500 images from the erased models using the prompt "A photo of a [*person name*]". For CI, we generate the same number of images by replacing [*person name*] with the special placeholder string $c_*$.

**NSFW content:** Introduced by Schramowski et al. (2023), the I2P dataset comprises 4703 unique prompts with corresponding seeds, which (to date) is the definitive benchmark for measuring the effectiveness of NSFW concept erasure. This process involves generating images using the prompts and seeds and subsequently utilizing NudeNet (Bedapudi, 2022) to classify the images into various nudity classes. The I2P benchmark is effective as its prompts do not necessarily contain words strictly related to nudity. Hence, an effective erasure method on this benchmark requires some degree of robustness to prompt selection. To evaluate each concept erasure method, we first used SD 1.4 to generate 4703 images using the I2P dataset. We used NudeNet to filter out 382 images with detected exposed body parts, on which we performed Concept Inversion. To measure how well the NSFW concept is recovered, we generated another 4703 images using the erased model by using the I2P prompts with the special placeholder string $c_*$ prepended, which are then evaluated by NudeNet.

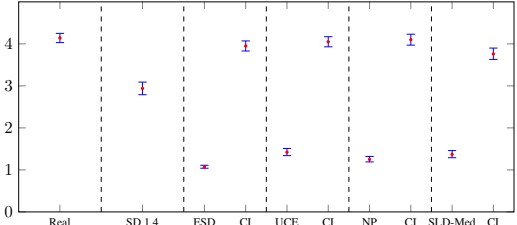

Figure 2: **Quantitative results of Concept Inversion for artistic concept:** Our human study ratings (with ± 95% confidence intervals) show that we can recover the erased artistic concept across all models. The CI Likert score is even higher than the images generated by SD 1.4.

### 4.2.1 ERASED STABLE DIFFUSION (ESD)

**Concept Erasure Method Details.** Gandikota et al. (2023a) fine-tune the pre-trained diffusion U-Net model weights to remove a specific style or concept. The authors reduce the probability of

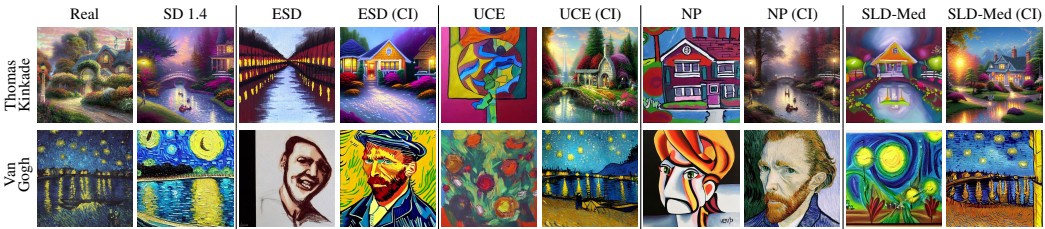

Figure 3: **Qualitative results of Concept Inversion for artistic concept:** Columns 4, 6, 8, and 10 demonstrate the effectiveness of concept erasure methods in not generating the targeted artistic concepts. However, we can still generate images of the erased styles using CI.

Table 1: **Quantitative results of Concept Inversion for object concept (Acc. % of erased model / Acc. % of CI):** Concept erasure methods can cleanly erase many object concepts from SD 1.4, evidenced by a significant drop in classification accuracy. Using Concept Inversion, we can generate images of the erased objects, which can be seen by an increase in average accuracy across all methods.

|  | SD 1.4 | ESD | UCE | NP | SLD-Med |
|---|---|---|---|---|---|
| cassette player | 6.4 | 0.2 / 6.2 | 0.0 / 2.8 | 4.0 / 9.4 | 1.0 / 2.4 |
| chain saw | 68.6 | 0.0 / 64.0 | 0.0 / 43.6 | 4.0 / 82.8 | 0.8 / 86.6 |
| church | 79.6 | 0.8 / 87.4 | 10.0 / 82.2 | 25.4 / 78.4 | 20.6 / 72.0 |
| english springer | 93.6 | 0.2 / 48.2 | 0.0 / 69.6 | 27.0 / 90.4 | 24.6 / 96.4 |
| french horn | 99.3 | 0.0 / 81.6 | 0.4 / 99.4 | 62.4 / 99.0 | 17.0 / 97.6 |
| garbage truck | 83.2 | 0.8 / 57.0 | 16.4 / 89.6 | 39.4 / 84.6 | 19.8 / 94.8 |
| gas pump | 76.6 | 0.0 / 73.8 | 0.0 / 73.0 | 18.0 / 79.6 | 12.8 / 75.6 |
| golf ball | 96.2 | 0.0 / 28.6 | 0.2 / 18.6 | 45.2 / 88.4 | 60.2 / 98.8 |
| parachute | 96.2 | 0.0 / 94.2 | 1.6 / 94.2 | 32.8 / 77.2 | 52.8 / 95.8 |
| tench | 79.6 | 0.3 / 59.7 | 0.0 / 20.6 | 27.6 / 72.6 | 20.6 / 75.4 |
| Average | 77.9 | 0.2 / 60.1 | 2.9 / 59.4 | 28.6 / 76.2 | 23.0 / 79.5 |

generating an image $x$ based on the likelihood described by the textual description of the concept, i.e. $\mathbb{P}_{\theta^*}(x) \propto \frac{\mathbb{P}_\theta(x)}{\mathbb{P}_\theta(c|x)^\eta}$, where $\theta^*$ is the updated weights of the diffusion model (U-Net), $\theta$ is the original weights, $\eta$ is a scale power factor, $c$ is the target concept to erase, and $\mathbb{P}(x)$ represents the distribution generated by the original model. Based on Tweedie's formula ( Efron (2011)) and the reparametrization trick ( Ho et al. (2020)), the authors derive a denoising prediction problem as $\epsilon_{\theta^*}(x_t, c, t) \leftarrow \epsilon(x_t, t) - \eta[\epsilon_\theta(x_t, c, t) - \epsilon_\theta(x_t, t)]$. By optimizing this equation, the fine-tuned model's conditional prediction is steered away from the erased concept when prompted with it. The authors propose two variants of ESD: ESD-$x$ and ESD-$u$, which finetune the cross-attentions and unconditional layers (non-cross-attention modules) respectively.

**Concept Inversion Method.**   We employ standard Textual Inversion on fine-tuned Stable Diffusion models from Gandikota et al. (2023a) to learn a new word embedding that corresponds to the concept of the training images. The authors provide pre-trained ESD-$x$ models for all 6 artistic concepts, the pre-trained ESD-$u$ model for NSFW content concept, and training scripts for object concepts. For ID concepts, we train our own ESD-$u$ models prior to CI.

### 4.2.2 UNIFIED CONCEPT EDITING (UCE)

**Concept Erasure Method Details.**   Latent diffusion models (Rombach et al., 2022) operate on low-dimensional embedding that is modeled with a U-Net generation network. The model incorporates conditioned textual information via embeddings derived from a language model. These embeddings are introduced into the system via cross-attention layers. Inspired by Orgad et al. (2023) and Meng et al. (2023), Gandikota et al. (2023b) edit the U-Net of Stable Diffusion models without training using a closed-form solution conditioned on cross-attention outputs. They update attention weights to induce targeted changes to the keys/values that correspond to specific text embeddings for a set of edited concepts, while minimizing changes to a set of preserved concepts.

**Concept Inversion Method.**   We employ standard Textual Inversion (Gal et al., 2023) on fine-tuned Stable Diffusion models from UCE (Gandikota et al., 2023b) to learn a new word embedding that corresponds to the (identity) concept of the training images. Gandikota et al. (2023b) provide training

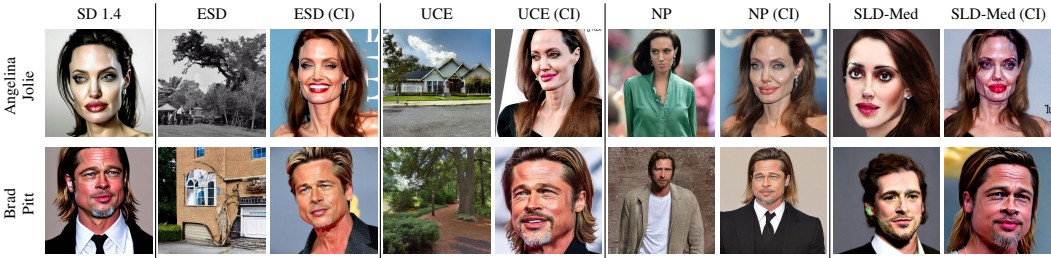

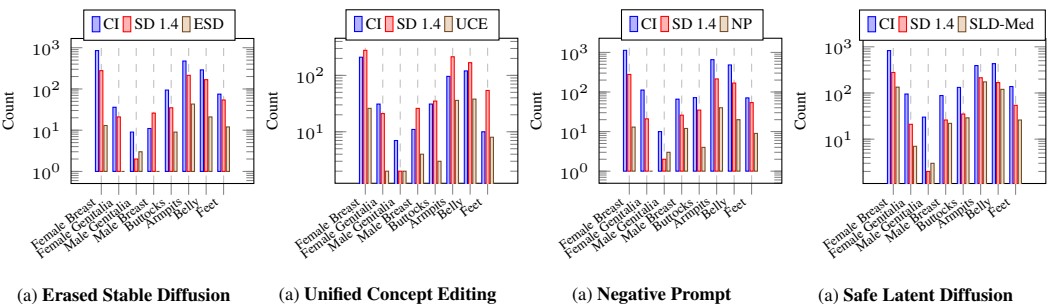

Figure 4: **Qualitative results of Concept Inversion for ID concept:** Columns 3, 5, 7, and 9 demonstrate the effectiveness of concept erasure methods in not generating Brad Pitt and Angelina Jolie. However, we can still generate images of the erased IDs using CI.

| (a) **Erased Stable Diffusion** | (a) **Unified Concept Editing** | (a) **Negative Prompt** | (a) **Safe Latent Diffusion** |

Figure 5: **Quantitative results of Concept Inversion for NSFW concept:** On average, the number of detected body parts from SD 1.4 and the erased models is 99.75 (across 4703 images) and 26.21 (across 4703 images and 7 erasure methods), respectively. Using CI, the average number of detected body parts is 170.93 across 7 methods.

scripts to reproduce their art style, object, and NSFW content concepts. For ID concepts, we adapt their publicly posted code to train our own models.

### 4.2.3 NEGATIVE PROMPT (NP)

**Concept Erasure Method Details.** Negative Prompt (NP) is a guiding inference technique used in the Stable Diffusion community (AUTOMATIC1111, 2022). Instead of updating the weights of the original model, it replaces the unconditional score with the score estimate conditioned on the erased concept in classifier-free guidance. Gandikota et al. (2023a) illustrate how NP can prevent the image generation of targeted artistic concepts.

**Concept Inversion Method.** In our experiments, vanilla Textual Inversion was not able to circumvent NP. We modify the objective function for Textual Inversion to:

$$v_* = \arg\min_v \mathbb{E}_{z \sim \mathcal{E}(x), c, \epsilon \sim \mathcal{N}(0,1), t} \Big[ \| (\epsilon_\theta(z_t, t) + \alpha(\epsilon_\theta(z_t, c, t) - \epsilon_\theta(z_t, t))$$
$$- (\epsilon_\theta(z_t, c, t) + \alpha(\epsilon_\theta(z_t, c_*, t) - \epsilon_\theta(z_t, c, t))) \|_2^2 \Big],$$

where $c$ is the target concept. Our method learns a word embedding associated with the special string $c_*$ such that the predicted noise from NP equals the true predicted noise using classifier-free guidance.

### 4.2.4 SAFE LATENT DIFFUSION (SLD)

**Concept Erasure Method Details.** Safe Latent Diffusion (Schramowski et al., 2023) is an inference guiding method that is a more sophisticated version of NP, where the second unconditional score term is replaced with a safety guidance term. Instead of being constant like NP, this term is dependent on: (1) the timestep, and (2) the distance between the conditional score of the given prompt and the conditional score of the target concept at that timestep. In particular, SLD modifies the score

estimates during inference as $\bar{\epsilon}_\theta(x_t, c, t) \leftarrow \epsilon_\theta(x_t, t) + \mu[\epsilon_\theta(x_t, c, t) - \epsilon_\theta(x_t, t) - \gamma(z_t, c, c_S)]$. We refer the reader to the Appendix C and the original work by Schramowski et al. (2023) for a more detailed explanation of $\gamma(z_t, c, c_S)$. Note that SLD does not modify the weights of the original diffusion models, but only adjusts the sampling process. By varying the hyperparameters of the safety guidance term, the authors propose 4 variants of SLD: SLD-Weak, SLD-Medium, SLD-Strong, and SLD-Max. A more potent variant of SLD yields greater success in erasing undesirable concepts.

**Concept Inversion Method.** In our experimentation, we encountered an issue akin to the one with Negative Prompt when applying vanilla Textual Inversion. Furthermore, the guidance term of SLD at timestep $t$ depends on that at the previous timestep. This recursive dependency implies that in order to calculate the guidance term within our inversion algorithm, it becomes necessary to keep a record of the preceding terms for optimization purposes. Given the high number of denoising steps involved, such a process could result in significant memory usage, presenting an efficiency problem. To address this issue, we propose a new strategy to perform Concept Inversion. Instead of having a constant safety guidance term, SLD requires storing all the guidance terms from step 1 to step $t-1$ to calculate the one at timestep $t$. Since doing so will be memory-intensive, we instead approximate it by calculating only a subset of guidance terms at evenly spaced timesteps between 1 and $t$. We can then learn a word embedding that counteracts the influence of the safety guidance term. The pseudocode for our CI scheme can be found in Appendix C.

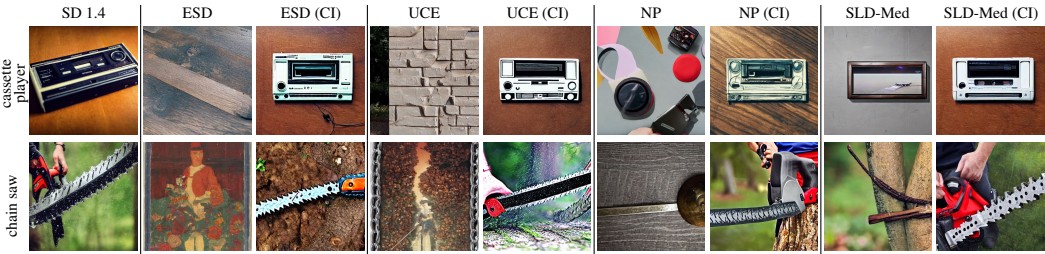

Figure 6: **Qualitative results of Concept Inversion for object concept:** Columns 3, 5, 7, and 9 demonstrate the effectiveness of concept erasure methods in not generating the targeted object concepts. However, we can still generate images of the object using CI. We refer the readers to the Appendix C for the complete results on all object classes.

Table 3: **Quantitative results of Concept Inversion for ID concept (Acc. % of erased model / Acc. % of CI):** Concept erasure methods can cleanly erase images of Brad Pitt and Angelina Jolie from SD 1.4, evidenced by a significant drop in classification accuracy. CI can recover images of the erased IDs, which can be seen by an increase in average accuracy across all methods.

|  | SD 1.4 | ESD | UCE | NP | SLD-Med |
|---|---|---|---|---|---|
| Brad Pitt | 90.2 | 0.0 / 61.2 | 0.0 / 59.4 | 43.2 / 71.4 | 4.8 / 71.8 |
| Angelina Jolie | 91.6 | 0.8 / 60.1 | 0.0 / 65.2 | 46.2 / 75.2 | 5.2 / 72.8 |
| Average | 90.9 | 0.4 / 60.7 | 0.0 / 62.3 | 44.7 / 73.2 | 5.0 / 72.3 |

## 4.3 RESULTS AND DISCUSSION

On the plus side, our experiments confirm that whenever the target concept is explicitly mentioned in the input prompt, all sevem concept erasure methods are effective. Therefore, these methods can indeed provide protection against obviously-offensive text inputs. We confirm this even for concept categories that the methods did not explore in their corresponding publications. However, on the minus side, all seven methods can be fully circumvented using our CI attacks. In other words, these erasure methods are only effective against their chosen inputs.

For artistic concepts, our human study in Figure 2 shows an average (across all methods) score of 1.31 on Likert rating on images generated by the erased model. This score expectedly increases to 3.85 when Concept Inversion is applied. Figure 3 displays images generated from the erased model and CI side-by-side, which shows that CI is effective in recovering the erased style. For object concepts, the average accuracy across all methods of the pre-trained classifier in predicting the erased concept

increases from 13.68 to 68.8 using our attack (Table 1). This is supported by qualitative results shown in Figure 6. For ID concepts, the average accuracy across all methods of the GIPHY detector increases from 12.52 to 67.13 in Table 3. For NSFW concepts, Figure 5 suggests that CI can recover the NSFW concept, which is shown by an increase from 26.2 to 170.93 in the average number of detected exposed body parts.

Among the 7 concept erasure methods, the most challenging one to circumvent is SLD. Our Concept Inversion technique manages to counteract the influence of SLD-Weak, SLD-Medium, and even SLD-Strong under most circumstances. Among the variants, SLD-Max proves to be the most challenging to circumvent. However, this variant comes with a drawback: it has the potential to entirely transform the visual semantics of the generated images. We provide several more results of variants of SLD in the Appendix C. In our proposed CI scheme for SLD, we observed that more GPU memory can give us better approximations of the safety guidance terms and therefore counteract their influence.

### 4.4 TRANSFERABILITY AND USEABILITY OF OUR ATTACKS

Intriguingly, we show that the learned special tokens derived from CI can be applied to the *original* SD 1.4 model to generate images of the erased concept. Figure 7 demonstrates our results. This lends further evidence that current concept erasure methods are merely suppressing the generation of the targeted concept when conditioned on the input embeddings corresponding to the concept text. However, there exist word embeddings that trigger the generation of the erased concept in SD 1.4 that also work on the erased model. This means that some degree of input filtering is occurring. Additionally, we provide evidence that the learned word embeddings through CI are useable in practice. Following Gal et al. (2023) and study the reconstruction effectiveness and editability of these embeddings. In particular, we generate two sets of images using CI with each concept erasure method: first, a set of generated images using the inverted concept; second, a set of generated images of the inverted concept in different scenes. We evaluate using CLIP (Radford et al., 2021) how well the inverted concepts are produced with the erased models, and how transferable are they to different scenes. In both cases, the erased models achieve performance similar to that of the original Stable-Diffusion model, pointing to the models are in principle still being able to produce the seemingly erased concepts.

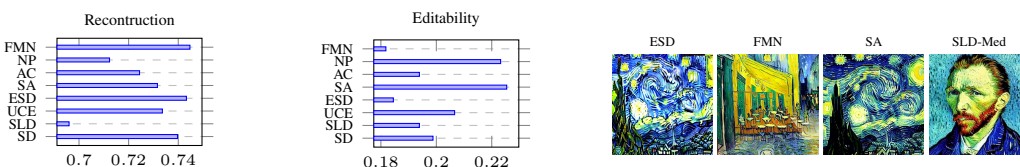

Figure 7: **(Left)** Reconstruction: A higher score indicates a better ability to replicate the provided concept. **(Middle)** Editability: A higher score indicates a better ability to modify the concepts using textual prompts. **(Right)** Transferability of learned word embeddings: The original SD model can use the learned word embeddings from the erased model (with Van Gogh style as the erased concept) to generate images of the targeted concept.

## 5 CONCLUSIONS

As text-to-image generative AI models continue to gain popularity and usage among the public, issues surrounding the ability to generate proprietary, sensitive, or unsafe images come to the fore. Numerous methods have been proposed in the recent past that claim to erase target concepts from trained generative models, and ostensibly make them safe(r) for public consumption. In this paper, we take a step back and scrutinize these claims. We show that post-hoc erasure methods have not excised the targeted concepts; fairly straightforward "attack" procedures can be used to design special prompts that regenerate the unsafe outputs.

As future work, it might be necessary to fundamentally analyze why the "input filtering" phenomenon seems to be occurring in all these recent methods, despite the diversity of algorithmic techniques involved in each of them. Such an understanding could facilitate the design of better methods that improve both the effectiveness and robustness of concept erasure.

ACKNOWLEDGMENTS

The authors were partially supported by the AI Research Institutes Program supported by NSF and USDA-NIFA under grant no. 2021-67021-35329. KM was supported by a DoE GAANN fellowship. NC was partially supported by the Israeli data science scholarship for outstanding postdoctoral fellows (VATAT).

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

APPENDIX

## A    LIMITATIONS AND DISCUSSION

The attack scenario described in Section 3 is closer to the white-box setting. However, despite limited practicality, our attacks offers valuable insights into the intrinsic limitations and behaviors of current concept erasure methods. Our experimental results suggest that input-filtering is occurring among concept erasure methods. Let $x$ be the observed data, i.e. the images containing the concept we wish to erase, and $\theta$ be the weights of our conditional generative model. From a likelihood-based perspective, we believe current erasure methods are minimizing the conditional probability $P_\theta(x|y)$ for only certain inputs y. In order to make the generative model more robust, we may want to directly minimize the marginal probability $P_\theta(x)$, which is unfortunately intractable to calculate in practice. One possible solution is to perform adversarial training and minimize probability conditioned on the adversarial inputs. Nevertheless, it is also crucial to maintain the overall generative performance of the model on unrelated concepts. For instance, Figure 10 demonstrates that the erased models that are more robust against Concept Inversion can create artifacts or generate images not aligned with the prompt.

## B    TRAINING DETAILS

For the training of Concept Inversion, we use 6 samples for art style concept, 30 samples for object concept, and 25 samples for ID concept. For all our Concept Inversion experiments, unless mentioned otherwise, we perform training on one A100 GPU with a batch size of 4, and a learning rate of $5e-03$. We optimize the word embedding for 1,000 steps while keeping the weights of the erased models frozen. The CI training procedure is the same across erasure methods and concepts, except for ID concepts we optimize for 5,000 steps, and for SLD we we train for 1,000 steps with batch size 1.

### B.1    ERASED STABLE DIFFUSION (ESD)

For artistic concepts and NSFW concept, we used the pre-trained models released by the authors. For object concepts, we trained ESD-$u$ models using the suggested parameters by the authors. For ID concepts, we trained ESD-$u$ models using the parameters for object concepts.

### B.2    UNIFIED CONCEPT EDITING (UCE)

For artistic concepts, object concepts, and NSFW concept, we trained the erased models using the training script provided by the author. For ID concepts, we used the same script without the preservation loss term.

### B.3    SELECTIVE AMNESIA (SA)

For artistic concepts, generated images with photorealism style as the surrogate dataset. We then applied SA to finetune the cross-attention layers of SD 1.4 for 200 epochs. We used the default parameters for training. For object concepts, we used a similar training procedure as above but used images of kangaroos (a class in ImageNet) as the surrogate dataset. Moreover, we finetuned the unconditional layers (non-cross-attention modules) of SD 1.4. For ID concepts and NSFW concept, we used the training script provided by the authors.

### B.4    ABLATING CONCEPTS (AC)

For artistic concepts, we used the training scripts provided by the authors to train the erased models. For object concepts, we also used the training script provided by the authors with the anchor distribution set to images of kangaroos. For ID concepts, we used the training script for object concepts and set the anchor distribution to images of middle-aged people. For NSFW concept, we used the training script for object concepts and set the anchor distribution to images of people wearing clothes.

## B.5 Forget-Me-Not (FMN)

For artistic concepts, we employed FMN to finetune the cross-attention layers for 5 training steps with a learning rate of $1e - 4$. The reference images we used are generated paintings with a photorealism style. For object concepts, we finetuned the unconditional layers (non-cross-attention modules) for 100 epochs with a learning rate of $2e - 06$. The reference images we used are generated photos of kangaroos. For ID concepts, we finetuned the unconditional layers for 120 epochs with a learning rate of $2e - 06$. The reference images we used are generated photos of middle-aged people. For NSFW concept, we finetuned unconditional layers for 100 epochs with a learning rate of $2e - 06$. The reference images we used are generated photos of people wearing clothes. Since the authors do not provide any training details, we tuned the number of epochs and learning rate until we observed effective erasure.

## B.6 Negative Prompt (NP)

To apply Negative Prompt to SD 1.4, we changed the $negative\_prompt$ argument to the concept we would like to erase during inference using codebase from Hugging Face.

## B.7 Safe Latent Diffusion (SLD)

We used to authors' public code to apply SLD to Stable Diffusion 1.4. During CI, we sampled 35 evenly spaced timesteps between 1 and 1000. The range between the smallest and largest timesteps is 90. We set guidance scale $s_S = 5000$, warm-up step $\delta = 0$, threshold $\lambda = 1.0$, momentum scale $s_m = 0.5$, and momentum beta $\beta_m = 0.7$. This is the hyperparameters for SLD-Max.

# C Results On Other Concept Erasure Methods

## C.1 Selective Amnesia (SA)

**Concept Erasure Method Details.** Heng & Soh (2023) pose concept erasure as a problem of continual learning, taking inspiration from Elastic Weight Consolidation (EWC) (Kirkpatrick et al., 2017) and Generative Replay (Shin et al., 2017). Consider a dataset $\mathcal{D}$ that can be partitioned as $\mathcal{D} = \mathcal{D}_f \cup \mathcal{D}_r = \{(x_f^{(n)}, c_f^{(n)})\}_{n=1}^{N_f} \cup \{(x_r^{(n)}, c_r^{(n)})\}_{n=1}^{N_r}$, where $\mathcal{D}_f$ is the data to forget and $\mathcal{D}_r$ is the data to remember. The underlying distribution of $\mathcal{D}$ is given by $p(x, c) = p(x|c)p(c)$. In the case of concept erasure, $D_f$ contains the images of the concept we would like to erase, and $D_r$ consists of images we want to preserve the model performance. They maximize the following objective function for concept erasure:

$$\mathcal{L} = -\mathbb{E}_{p(x|c)p(c_f)}[\log p(x|\theta^*, c)] - \lambda \sum_i \frac{F_i}{2}(\theta_i - \theta_i^*)^2 + \mathbb{E}_{p(x|c)p(x_r)}[\log p(x|\theta^*, c)], \quad (2)$$

where $F$ is the Fisher information matrix and the third term is a generative replay term to prevent model degradation on samples that do not contain the erased concept. In practice, the authors optimize Eq. 2 by substituting the likelihood terms with the standard ELBOs. Moreover, they observe that directly minimizing the ELBO can lead to poor results. Hence, they propose to *maximize* the log-likelihood of a surrogate distribution of the concept to forget, $q(x|c_f) \neq p(x|c_f)$. This is done by replacing $-\mathbb{E}_{p(x|c)p(c_f)}[\log p(x|\theta^*, c)]$ with $\mathbb{E}_{q(x|c)p(c_f)}[\log p(x|\theta^*, c)]$. Intuitively, this fine-tuning will result in a generative model that will produce images according to the surrogate distribution when conditioned on $c_f$.

**Concept Inversion Method.** We employ standard Textual Inversion (Gal et al., 2023) on fine-tuned Stable Diffusion models from Heng & Soh (2023) to learn a new word embedding that corresponds to the concept of the training images. The authors provide training scripts for Brad Pitt, Angelina Jolie, and NSFW content concepts. In particular, they use images of clowns and middle-aged people as the surrogate dataset for ID concepts, and images of people wearing clothes for NSFW content concept. For other concepts, we train our own models by appropriately modifying their training scripts prior to CI.

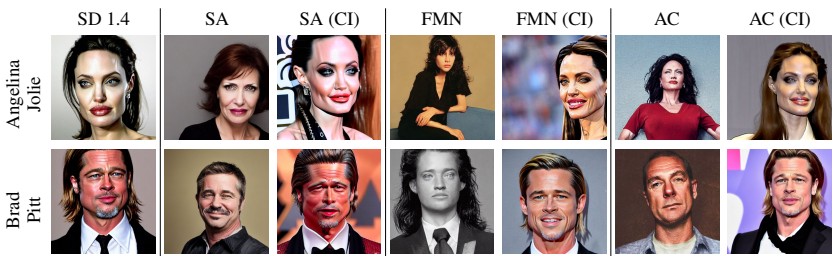

Figure 8: **Concept Inversion (CI) on concept erasure methods for IDs.** Columns 3, 5, and 7 demonstrate the effectiveness of concept erasure methods in not generating Brad Pitt and Angelina Jolie. However, we can still generate images of the erased IDs using CI.

**Experimental Results.** For artistic concepts, our human study in Figure 2 shows an average of 1.12 Likert score on images generated by the erased models. This score is increased to 3.55 when Concept Inversion is applied. Qualitative results in Figure 13 demonstrate that CI can recover the erased style. When evaluated on object concepts, Table 8 indicates that the average accuracy of the pre-trained classifier increases from 10.6 to 40.3. With respect to ID concepts, the accuracy of GIPHY detector increases from 4.8 to 67.1 in Table 6. Figure 8 shows that CI can still generate images of Brad Pitt and Angelina Jolie. In the context of NSFW content, the average number of exposed body parts detected increases from 2.5 to 5 in Figure 11. While CI is not effective in recovering the NSFW concept, we observe that the performance of the erased model on unrelated concepts is affected.

## C.2 FORGET-ME-NOT (FMN)

**Concept Erasure Method Details.** Zhang et al. (2023) propose fine-tuning the cross-attention layers of Stable Diffusion's U-Net to map the erased concept to that of a set of reference images. The authors first locate the word embeddings associated with the forgetting concept. This can be done by tokenizing a pre-defined prompt or through Textual Inversion. They then compute the attention maps between the input features and these embeddings, and minimize the Frobenius norm of attention maps and backpropagate the network. Algorithm 1 describes the concept erasure process.

---

**Algorithm 1** Forget-Me-Not on diffuser

---

**Require:** Context embeddings $\mathcal{C}$ containing the forgetting concept, embedding locations $\mathcal{N}$ of the forgetting concept, reference images $\mathcal{R}$ of the forgetting concept, diffuser $G_\theta$, diffusion step $T$.
    **repeat**
        $t \sim \text{Uniform}([1...T]); \epsilon \sim \mathcal{N}(0, I)$
        $r_i \sim \mathcal{R}; e_j \sim \mathcal{C}; n_j \sim \mathcal{N}$
        $x_0 \leftarrow r_i$
        $x_t \leftarrow \sqrt{\overline{\alpha}_t}x_0 + \sqrt{1 - \overline{\alpha}_t}\epsilon$         $\triangleright \overline{\alpha}_t$ : noise variance schedule
        $x_{t-1}, A_t \leftarrow G_\theta(x_t, e_j, t)$         $\triangleright A_t$ : all attention maps
        $\mathcal{L} \leftarrow \sum_{a_t \in A_t} \|a_t^{[n_j]}\|^2$         $\triangleright \mathcal{L}$ : attention resteering loss
        Update $\theta$ by descending its stochastic gradient $\nabla_\theta \mathcal{L}$
    **until** Concept forgotten

---

**Concept Inversion Method.** We employ standard Textual Inversion Gal et al. (2023) on fine-tuned Stable Diffusion models from Zhang et al. (2023) to learn a new word embedding that corresponds to the concept of the training images. Zhang et al. (2023) only provides training scripts for ID concepts. Hence, we have to train our models on other concepts using their public code prior to CI.

**Experimental Results.** For artistic concepts, our human study in Figure 2 shows an average of 1.43 Likert score on images generated by the erased models. This score is increased to 3.66 when Concept Inversion is applied. Our qualitative results in Figure 13 also shows that CI can reconstruct the erased artistic concept. When evaluated on object concepts, Table 8 indicates that the average accuracy of the pre-trained classifier increases from 3.9 to 44.6. In cases where CI is not as effective (chain saw, church, english springer), we observe that the erased model has poor performance in generating unrelated images. With respect to ID concepts, the accuracy of GIPHY detector increases

Table 5: **Quantitative results of Concept Inversion for ID concept (Acc. % of erased model / Acc. % of CI):** Concept erasure methods can cleanly erase images of Brad Pitt and Angelina Jolie from the original SD 1.4 model, evidenced by a significant drop in classification accuracy. Using Concept Inversion, we can generate images of the erased IDs, which can be seen by an increase in average accuracy across all methods.

|  | SD 1.4 | ESD | FMN | UCE | AC | NP | SLD-Med | SA |
|---|---|---|---|---|---|---|---|---|
| Brad Pitt | 90.2 | 0.0 / 61.2 | 0.6 / 52.8 | 0.0 / 59.4 | 3.2 / 73.6 | 43.2 / 71.4 | 4.8 / 71.8 | 0.0 / 66.6 |
| Angelina Jolie | 91.6 | 0.8 / 60.1 | 0.0 / 41.2 | 0.0 / 65.2 | 0.6 / 79.6 | 46.2 / 75.2 | 5.2 / 72.8 | 9.6 / 67.7 |
| Average | 90.9 | 0.4 / 60.7 | 0.3 / 47.0 | 0.0 / 62.3 | 1.9 / 76.6 | 44.7 / 73.2 | 5.0 / 72.3 | 4.8 / 67.1 |

from 0.3 to 47.0 in Table 6. Additionally, in the context of NSFW content, the average number of exposed body parts detected increases from 1.88 to 52.13 in Figure 11.

## C.3 ABLATING CONCEPTS (AC)

**Concept Erasure Method Details.** Kumari et al. (2023) perform concept erasure by overwriting the target concept with an anchor weight, which can be a superset or a similar concept. The authors propose two variants to erase the target concept, namely Model-based concept ablation and Noise-based concept ablation. In the former method, the authors fine-tune the pre-trained Stable Diffusion U-Net model by minimizing the following objective function:

$$\arg \min_{\theta^*} \mathbb{E}_{z \sim \mathcal{E}, z^* \sim \mathcal{E}(x^*), c, c^*, \epsilon \sim \mathcal{N}(0,1), t} \left[ w_t \| \epsilon_{\theta^*}(z_t, c, t).sg() - \epsilon_{\theta^*}(z_t^*, c^*, t) \|_2^2 \right].$$

where $w_t$ is a time-dependent weight, $\theta^*$ is initialized with the pre-trained weight, $x^*$ is the (generated) images with the anchor concept $c^*$, and $.sg()$ is the stop-gradient operation. For the second variant, the authors redefine the ground truth text-image pairs as <*a target concept text prompt, image of the anchor concept*>. The authors then fine-tune the model on the redefined pairs with the standard diffusion training loss. In addition, they add an optional standard diffusion loss term on the anchor concept image and corresponding texts as a regularization as the target text prompt can consist of the anchor concept. In both variants, the authors propose to fine-tune on different parts of Stable Diffusion: (1) cross-attention, (2) embedding: the text embedding the text transformer, (2) full weights: all parameters of U-Net.

**Concept Inversion Method.** We employ standard Textual Inversion Gal et al. (2023) on fine-tuned Stable Diffusion models from AC ( Kumari et al. (2023)) to learn a new word embedding that corresponds to the (identity) concept of the training images. Kumari et al. (2023) provide training scripts for art style and object concepts. Consequently, we extend their public code to erase the remaining concepts.

**Experimental Results.** For artistic concepts, our human study in Figure 2 shows an average of 1.47 Likert score on images generated by the erased models. This score is increased to 3.84 when Concept Inversion is applied. When evaluated on object concepts, Table 8 indicates that the average accuracy of the pre-trained classifier increases from 0.04 to 45.5. With respect to ID concepts, the accuracy of GIPHY detector increases from 1.9 to 76.6 in Table 6. Additionally, in the context of NSFW content, the average number of exposed body parts detected increases from 12.5 to 252.63 in Figure 11.

## C.4 SAFE LATENT DIFFUSION (SLD)

**Concept Erasure Method Details.** Safe Latent Diffusion Schramowski et al. (2023) is an inference guiding method, in which during inference the score estimates for the $x-$prediction are modified as:

$$\bar{\epsilon}_\theta(x_t, c, t) \leftarrow \epsilon_\theta(x_t, t) + \mu[\epsilon_\theta(x_t, c, t) - \epsilon_\theta(x_t, t) - \gamma(z_t, c, c_S)] \tag{3}$$

Where $c_S$ is the concept we would like to steer the generated images away from, $\gamma$ is the safety guidance term $\gamma(z_t, c, c_S) = \beta(c, c_S; s_S, \lambda)(\epsilon_\theta(x_t, c, t) - \epsilon_\theta(x_t, t))$, and $\beta$ applies a guidance scale $s_S$ element-wise by considering those dimensions of the prompt conditioned estimate that would guide the generation process toward the erased concept: $\beta(c, c_S; s_S, \lambda) =$

$$\begin{cases} \max\{1, |s_S(\theta_\epsilon(x_t, c_t) - \epsilon(x_t, c_S, t)|\}, & \text{where } \epsilon_\theta(x_t, c, t) \ominus \epsilon_\theta(x_t, c_S, t) \leq \lambda, \\ 0, & \text{otherwise.} \end{cases}$$

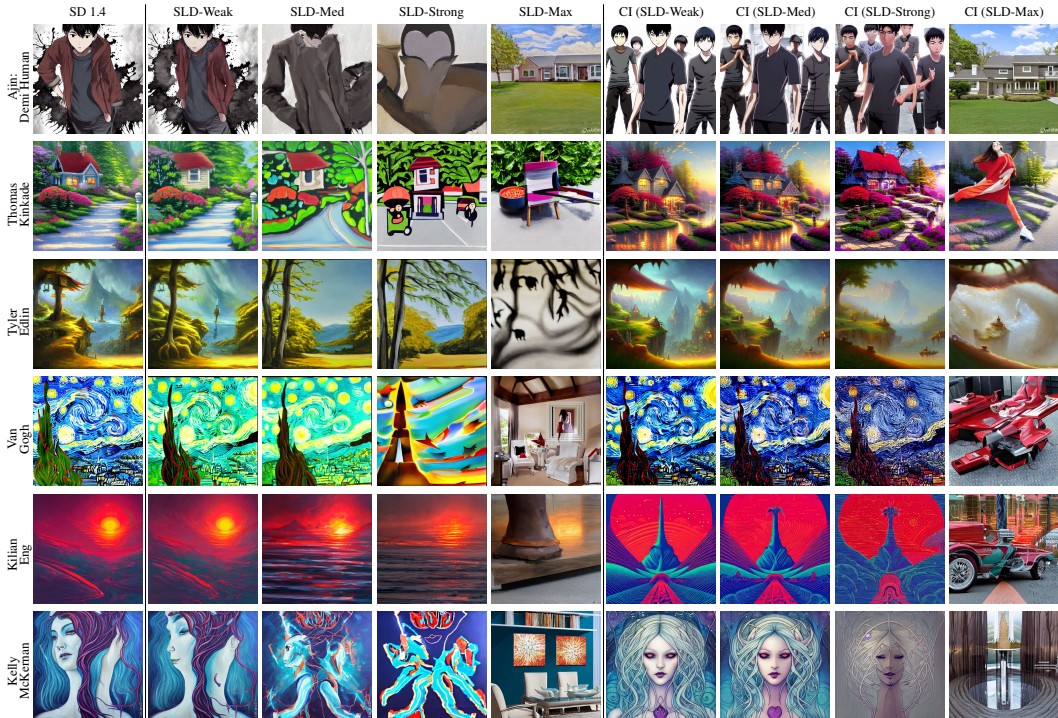

Figure 9: **Concept Inversion (CI) on SLD for art concept.** Columns 2 to 5 demonstrate the effectiveness of erasing artistic styles for each SLD variant. Concept Inversion can recover the style most consistently for SLD-Weak and SLD-Strong. In most cases, we can observe recovery for even SLD-Strong.

Larger threshold $\lambda$ and $s_S$ lead to a more substantial shift away from prompt text and in the opposite direction of the erased concept. Moreover, to facilitate a balance between removal of undesired content from the generated images and minimizing the changes introduced, the authors introduce two modifications. First, they accelerate guidance over time by utilizing a momentum term $v_t$ for the safety guidance $\gamma$. In particular, $\gamma_t$, denoting $\gamma(z_t, c, c_S)$, is defined as:

$$\gamma_t = \beta(c, c_S; s_S, \lambda)(\epsilon_\theta(x_t, c, t) - \epsilon_\theta(x_t, t)) + s_m v_t$$

with $s_m \in [0, 1]$ and $v_{t+1} = \zeta_m v_t + (1 - \zeta_m)\gamma_t$. Here, $v_0 = \mathbf{0}$ and $\beta_m \in [0, 1)$, with larger $\zeta_m$ resulting in less volatile changes in momentum. Note that SLD does not modify the weights of the original diffusion models, but only adjusts the sampling process. By varying the hyperparameters, the authors propose 4 variants of SLD: SLD-Weak, SLD-Medium, SLD-Strong, and SLD-Max.

**Concept Inversion Method.** The pseudocode for our CI method on SLD is shown in Alg. 2.

---

**Algorithm 2** Concept Inversion for SLD

---
**Require:** Concept placeholder string $c_*$, erased concept $c$, number of iterations $N$
    **for** $i = 1$ to $N$ **do**
        Sample $x_0$ from training data. Randomly select $m \geq 1$ and $n \leq T$
        $\epsilon \sim \mathcal{N}(0, 1)$; $z_0 \leftarrow \mathcal{E}(x_0)$; $\gamma(z_0, m, c_S) \leftarrow \mathbf{0}$; $v_m \leftarrow \mathbf{0}$
        **for** $t = m$ to $n$ by k **do**         ▷ iterating $t$ from $m$ to $k$ with an increment of $k$
            $\mathcal{L} \leftarrow \Big[\|(\epsilon_\theta(z_t, t) + \alpha(\epsilon_\theta(z_t, c_S, t) - \epsilon_\theta(z_t, t)) - (\epsilon_\theta(x_t, t) + \mu[\epsilon_\theta(x_t, c_*, t) - \epsilon_\theta(x_t, t) -$
$\gamma(z_t, c, c_S)])\|_2^2\Big]$
            $v_{t+1} \leftarrow \zeta_m v_t + (1 - \zeta_m)\gamma_t$
            $\gamma_{t+1} \leftarrow \beta(c, c_S; s_S, \lambda)(\epsilon_\theta(x_t, c, t) - \epsilon_\theta(x_t, t)) + s_m v_t$
            Update $v_*$ by descending its stochastic gradient $\nabla_{v_*}\mathcal{L}$
        **end for**
    **end for**

---

## C.5 FAILURE CASES FOR CI

In our experiments, we observed a small handful of cases where CI was not able to recover the erased concepts. However, as shown in Figure 10, these models can have poor image generation quality even on prompts not related to the erased concepts. Hence, this shows that while they are more robust to CI, such models are not very useful in real-world applications.

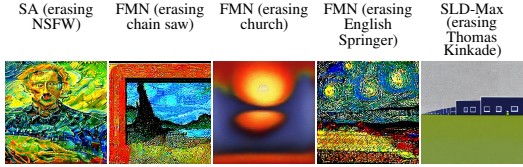

Figure 10: **Samples from erased models where CI struggles.** We generate images using the prompt "A painting in the style of Van Gogh". The output images do not contain the targeted artistic style and contain artifacts.

Table 7: **Quantitative results of Concept Inversion (CI) on concept erasure methods for objects. (Acc. % of erased model / Acc. % of CI)**. Concept erasure methods can cleanly erase many object concepts from a model when the object is mentioned in the prompt, evidenced by a significant drop in classification accuracy. Using Concept Inversion, we can generate images of the erased objects, which can be seen by an increase in average accuracy across all methods.

| | SD 1.4 | ESD | FMN | UCE | AC | NP | SLD-Med | SA |
|---|---|---|---|---|---|---|---|---|
| cassette player | 6.4 | 0.2 / 6.2 | 0.2 / 8.8 | 0.0 / 2.8 | 0.0 / 4.2 | 4.0 / 9.4 | 1.0 / 2.4 | 0.6 / 6.2 |
| chain saw | 68.6 | 0.0 / 64.0 | 0.0 / 0.2 | 0.0 / 43.6 | 0.0 / 17.8 | 4.0 / 82.8 | 0.8 / 86.6 | 0.0 / 2.0 |
| church | 79.6 | 0.8 / 87.4 | 0.0 / 0.0 | 10.0 / 82.2 | 0.4 / 72.6 | 25.4 / 78.4 | 20.6 / 72.0 | 56.2 / 65.6 |
| english springer | 93.6 | 0.2 / 48.2 | 0.0 / 0.0 | 0.0 / 69.6 | 0.0 / 32.6 | 27.0 / 90.4 | 24.6 / 96.4 | 0.0 / 8.2 |
| french horn | 99.3 | 0.0 / 81.6 | 0.0 / 59.0 | 0.4 / 99.4 | 0.0 / 66.6 | 62.4 / 99.0 | 17.0 / 97.6 | 0.2 / 87.0 |
| garbage truck | 83.2 | 0.8 / 57.0 | 6.4 / 69.6 | 16.4 / 89.6 | 0.0 / 79.4 | 39.4 / 84.6 | 19.8 / 94.8 | 12.6 / 35.4 |
| gas pump | 76.6 | 0.0 / 73.8 | 7.8 / 80.4 | 0.0 / 73.0 | 0.0 / 31.2 | 18.0 / 79.6 | 12.8 / 75.6 | 0.6 / 54.8 |
| golf ball | 96.2 | 0.0 / 28.6 | 22.6 / 74.4 | 0.2 / 18.6 | 0.0 / 28.4 | 45.2 / 88.4 | 60.2 / 98.8 | 3.0 / 49.0 |
| parachute | 96.2 | 0.0 / 94.2 | 2.0 / 93.4 | 1.6 / 94.2 | 0.0 / 92.4 | 32.8 / 77.2 | 52.8 / 95.8 | 22.6 / 78.6 |
| tench | 79.6 | 0.3 / 59.7 | 0.4 / 60.6 | 0.0 / 20.6 | 0.0 / 29.4 | 27.6 / 72.6 | 20.6 / 75.4 | 10.2 / 16.0 |
| Average | 77.9 | 0.2 / 60.1 | 3.9 / 44.6 | 2.9 / 59.4 | 0.04 / 45.5 | 28.6 / 76.2 | 23.0 / 79.5 | 10.6 / 40.3 |

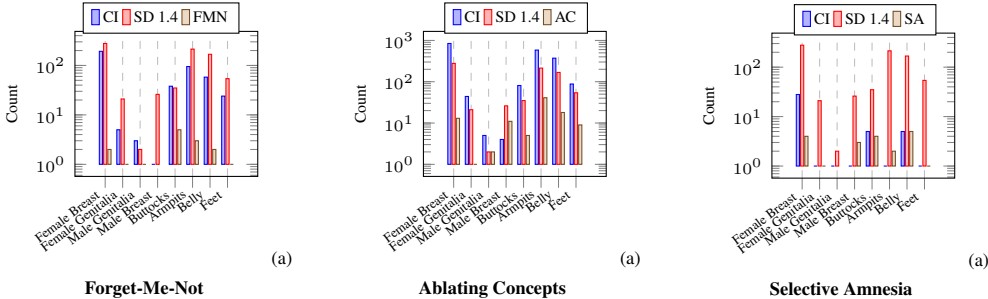

Figure 11: **Quantitative results of Concept Inversion for NSFW concept:** On average, the number of detected body parts from SD 1.4 and the erased models is 99.75 (across 4703 images) and 26.21 (across 4703 images and 7 erasure methods), respectively. Using CI, the average number of detected body parts is 170.93 across 7 methods.

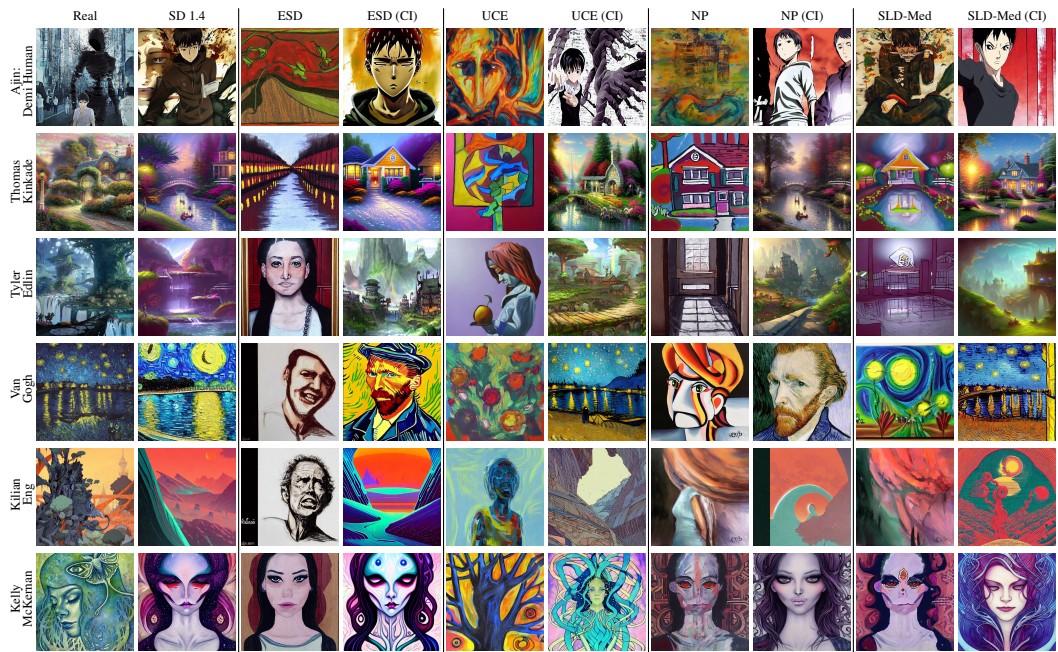

Figure 12: **Qualitative results of Concept Inversion on Erased Stable Diffusion, Unified Concept Editing, Negative Prompt, and Safe Latent Diffusion for artistic concept:** Columns 4, 6, 8, and 10 demonstrate the effectiveness of concept erasure methods in not generating the targeted artistic concepts. However, we can still generate images of the erased styles using CI.

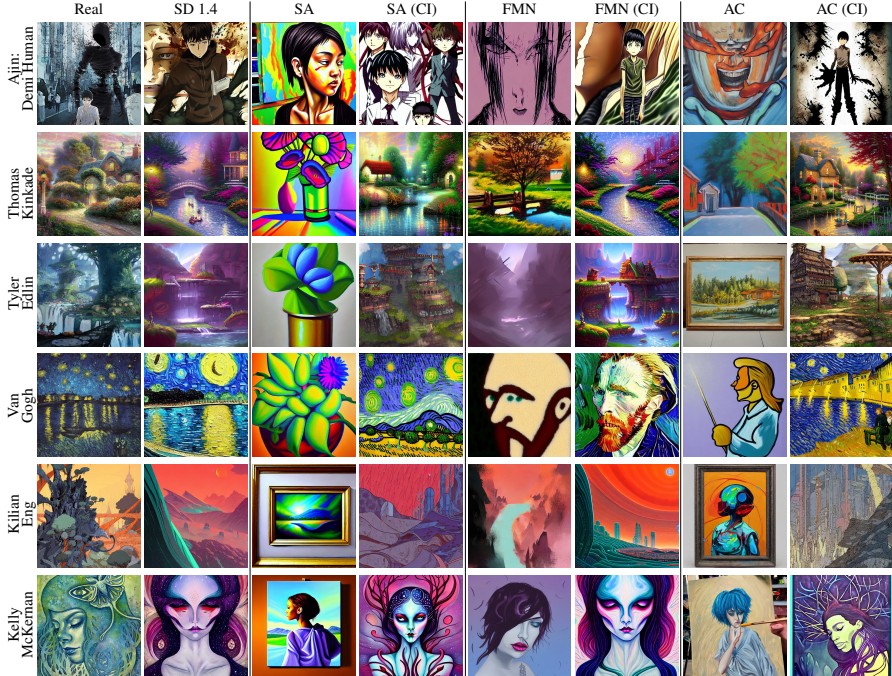

Figure 13: **Qualitative results of Concept Inversion on Selective Amnesia, Forget-Me-Not, and Ablating Concepts for artistic concept:** Columns 4, 6, and 8 demonstrate the effectiveness of concept erasure methods in not generating the targeted artistic concepts. However, we can still generate images of the erased styles using CI.

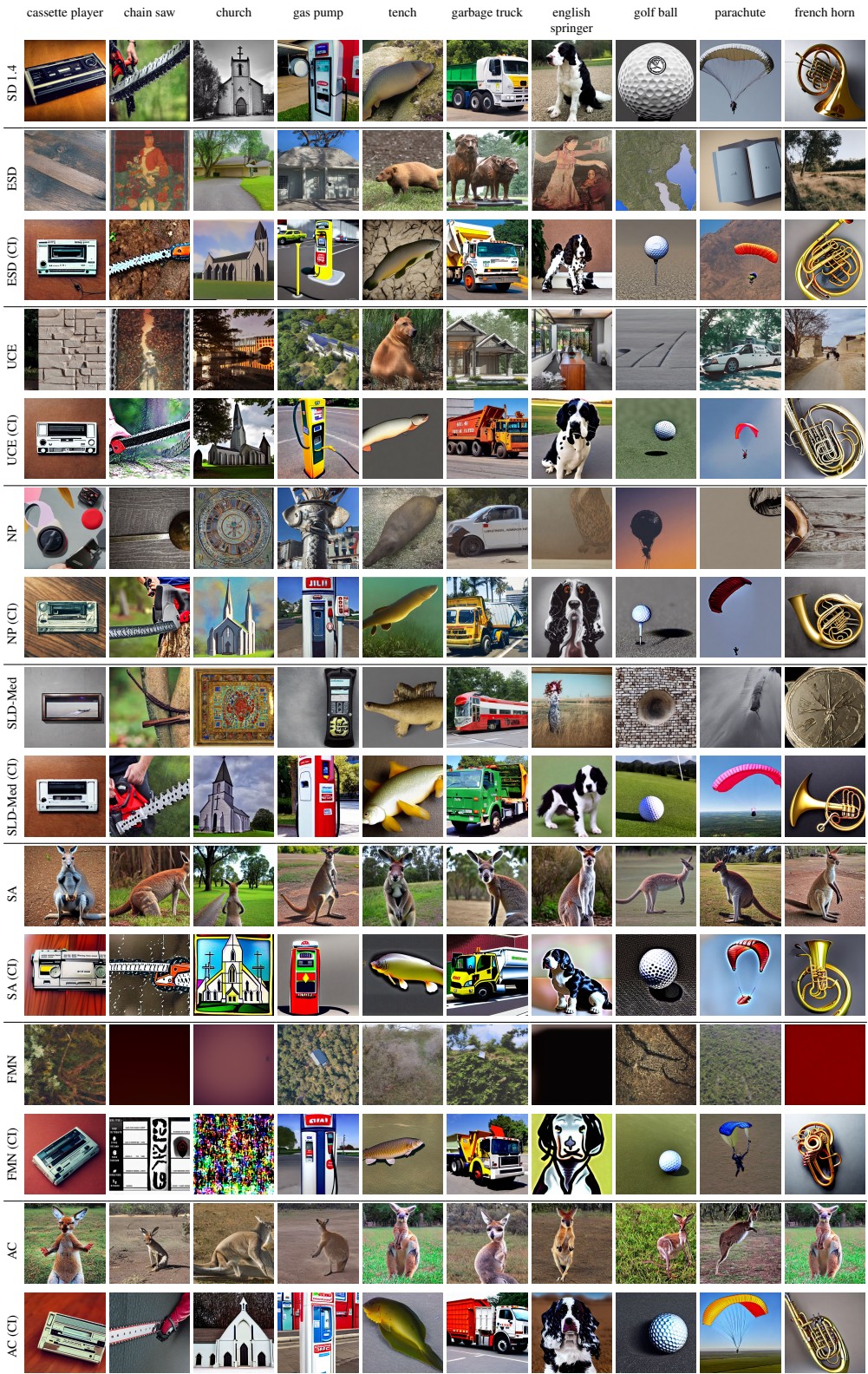

Figure 14: **Concept Inversion (CI) on concept erasure methods for objects**. Odd rows (except 1) demonstrate the effectiveness of erasure methods in erasing object concepts. However, Concept Inversion can recover the objects.

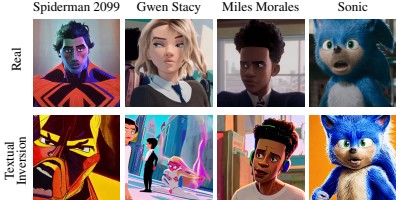 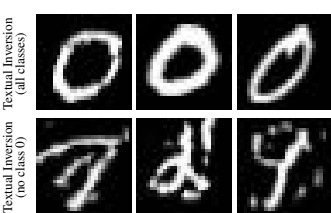

Figure 15: **(Left)** Qualitative results of Textual Inversion on 4 film character concepts: TI works significantly worse for concepts not abundant in the LAION 5-B ( Schuhmann et al. (2022)) such as Spiderman 2099 and Gwen Stacy compared to Miles Morales and Sonic, both of which we observed many duplicates. Refer to the Appendix D to see how we check for existence of these concepts in LAION 5-B. **(Right)** Qualitative results of Textual Inversion on MNIST: Textual Inversion works worse when excluding class 0 from the training dataset.

## D  DETAILS: CHECKING FOR EXISTENCE OF CONCEPTS

To check for the concepts' existence in LAION-5B, we utilize CLIP Retrieval ( Beaumont (2022)) to search for images nearest to the concept image in the CLIP image embedding space. Our searches indicate that the images of Sonic and Miles Morales appear a lot with multiple duplicates, while images of Gwen Stacy and Spiderman 2099 appear few to none.

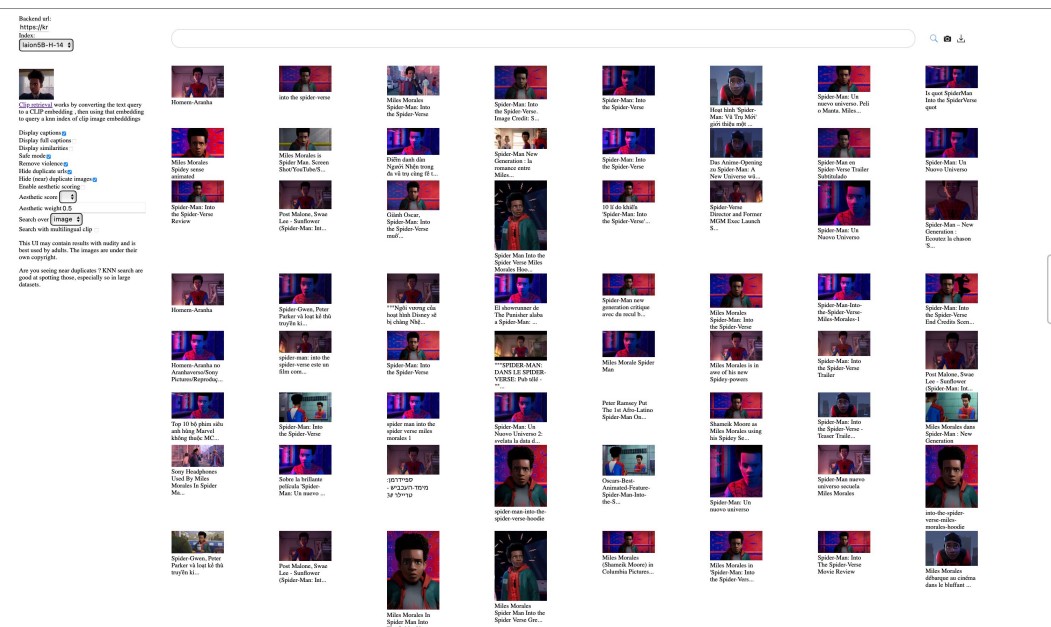

Figure 16: Images of the character Miles Morales appear with many duplicates in our search using `https://rom1504.github.io/clip-retrieval`.

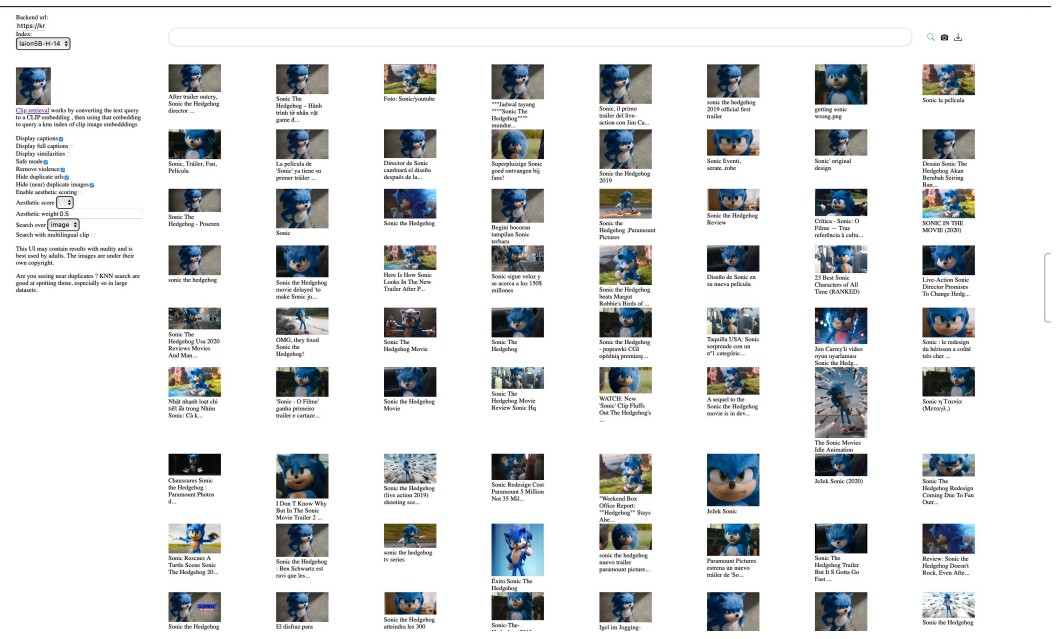

Figure 17: Images of the character Sonic appear with many duplicates in our search using `https://rom1504.github.io/clip-retrieval`.

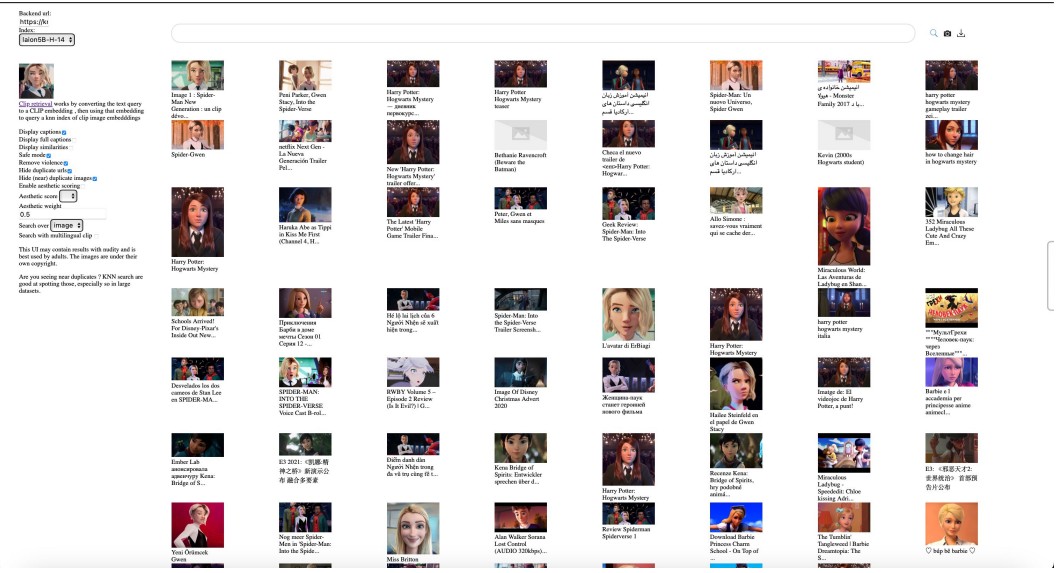

Figure 18: Images of the character Gwen Stacy appear only few in our search using `https://rom1504.github.io/clip-retrieval`.

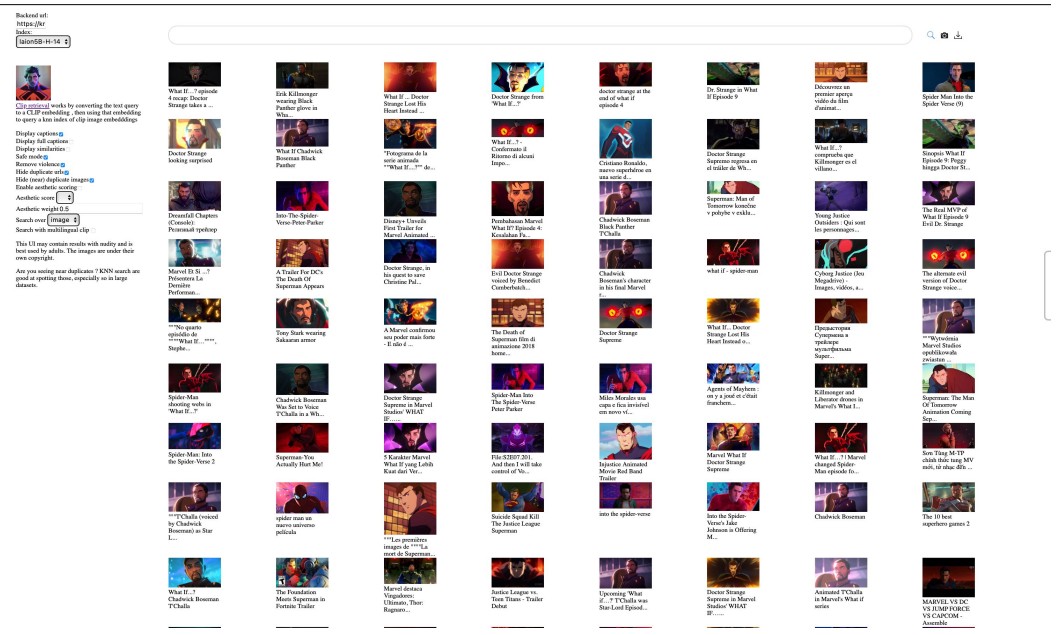

Figure 19: Images of the character Spiderman 2099 does not appear in our search using `https://rom1504.github.io/clip-retrieval`. This is reasonable since the character was introduced in a film released after the release of LAION-5B.

# E  ADDITIONAL QUALITATIVE RESULTS

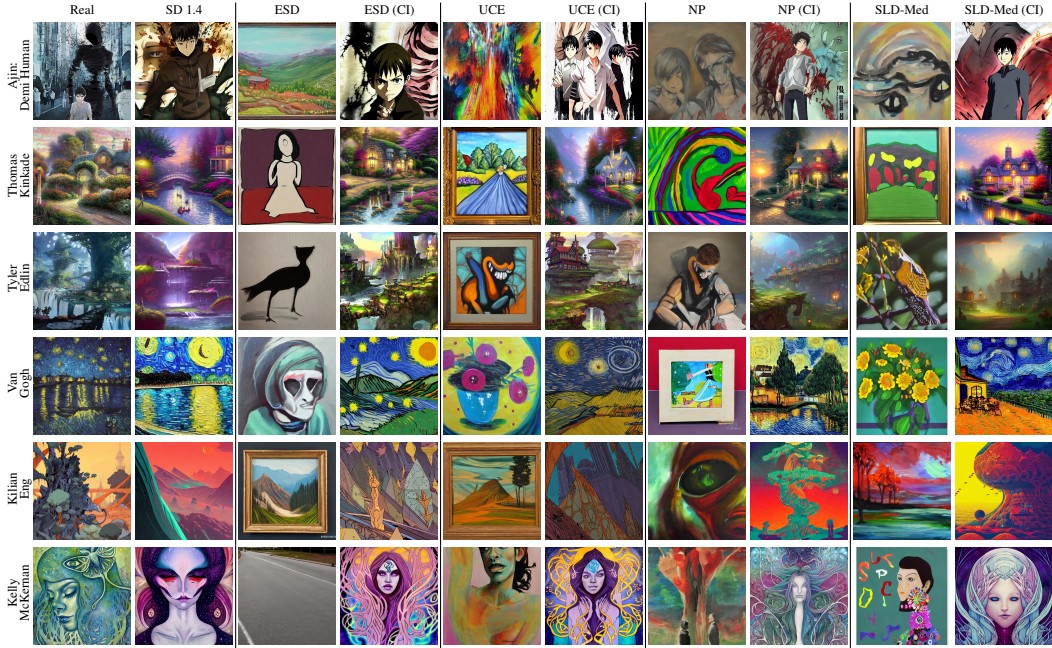

Figure 20: **Qualitative results of Concept Inversion on Erased Stable Diffusion, Unified Concept Editing, Negative Prompt, and Safe Latent Diffusion for artistic concept:** Columns 4, 6, 8, and 10 demonstrate the effectiveness of concept erasure methods in not generating the targeted artistic concepts. However, we can still generate images of the erased styles using CI.

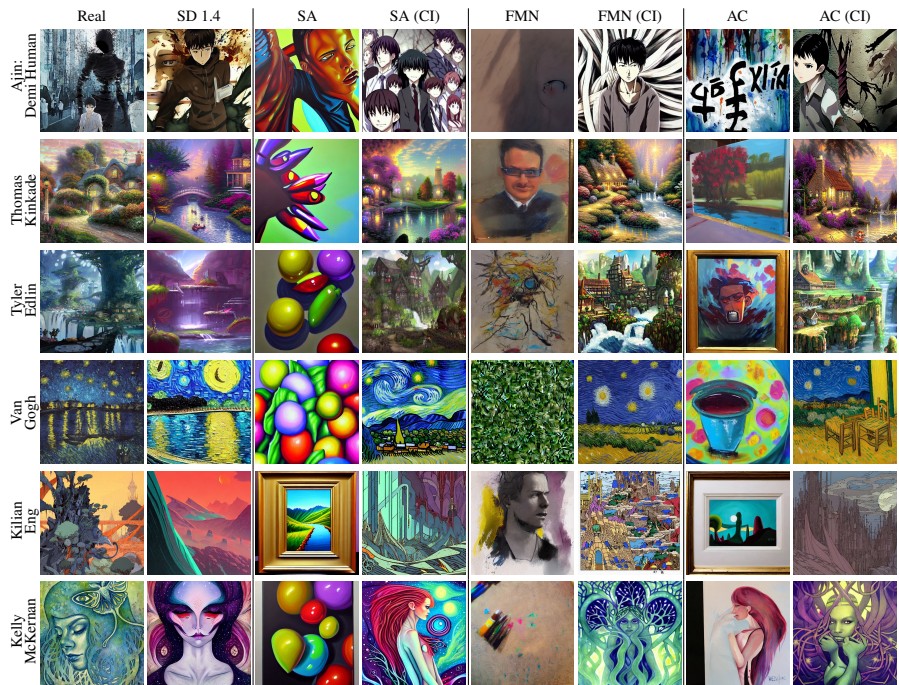

Figure 21: **Qualitative results of Concept Inversion on Selective Amnesia, Forget-Me-Not, and Ablating Concepts for artistic concept:** Columns 4, 6, and 8 demonstrate the effectiveness of concept erasure methods in not generating the targeted artistic concepts. However, we can still generate images of the erased styles using CI.

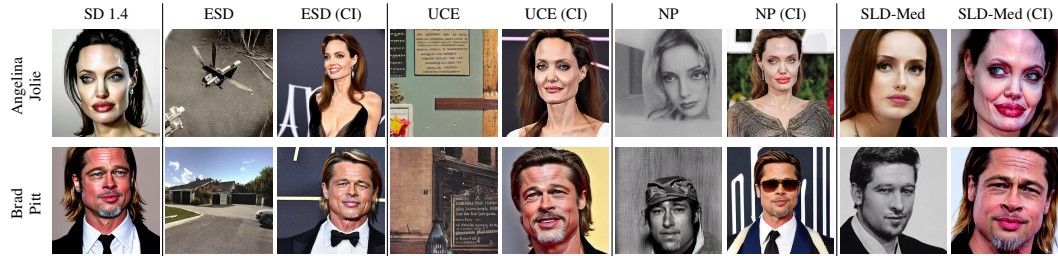

Figure 22: **Qualitative results of Concept Inversion for ID concept:** Columns 3, 5, 7, and 9 demonstrate the effectiveness of concept erasure methods in not generating Brad Pitt and Angelina Jolie. However, we can still generate images of the erased IDs using CI.

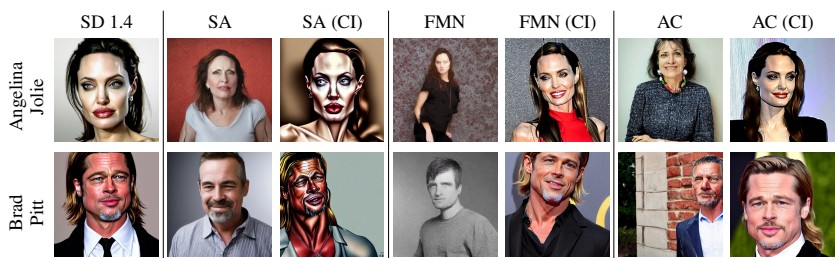

Figure 23: **Concept Inversion (CI) on concept erasure methods for IDs.** Columns 3, 5, and 7 demonstrate the effectiveness of concept erasure methods in not generating Brad Pitt and Angelina Jolie. However, we can still generate images of the erased IDs using CI.

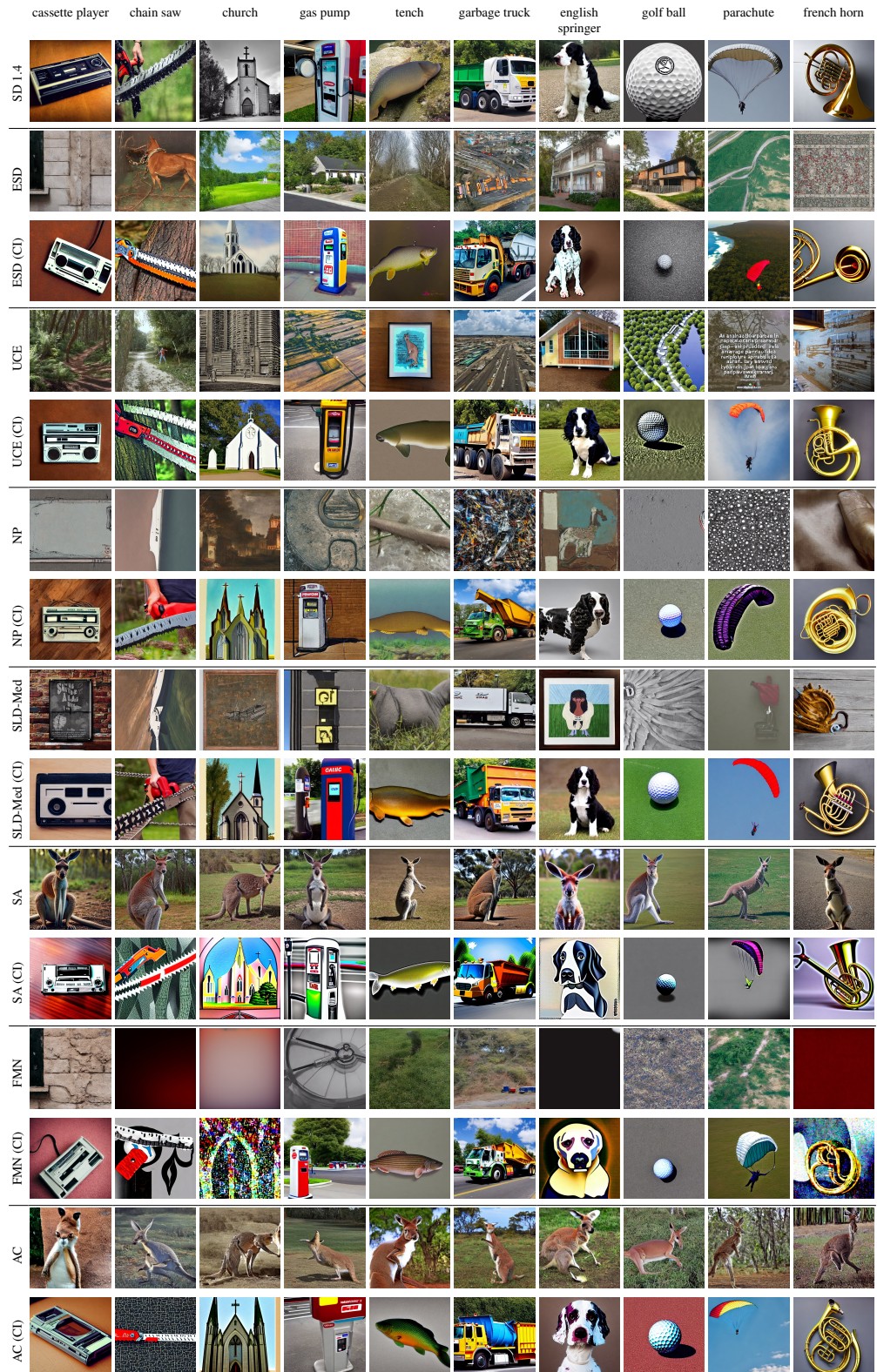

Figure 24: **Concept Inversion (CI) on concept erasure methods for objects**. Odd rows (except 1) demonstrate the effectiveness of erasure methods in erasing object concepts. However, Concept Inversion can recover the objects.

# F    ADDITIONAL RESULTS FOR STABLE DIFFUSION 2.0

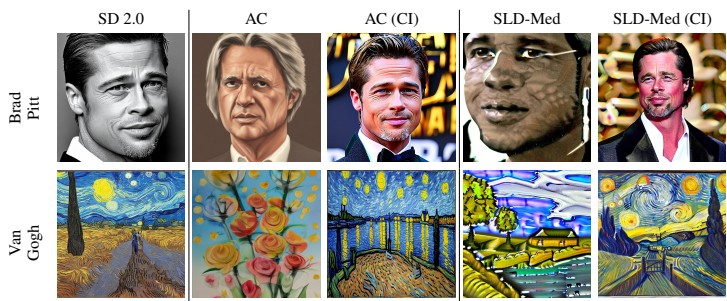

Figure 25: **Concept Inversion (CI) on concept erasure methods for IDs and art styles on SD 2.0.** For Brad Pitt concept, the classification accuracy is 96.6% for SD 2.0. This accuracy drops to 17.4% and 28.2% when AC ( Kumari et al. (2023)) and SLD ( Schramowski et al. (2023)) are applied, respectively. When CI is applied, the accuracy increases to 89.8% and 54.4%.

