# OpenReview forum: "Circumventing Concept Erasure Methods For Text-To-Image Generative Models"
_ICLR.cc/2024/Conference — ICLR 2024 poster_

### Official Review · Reviewer_h1Qn · 2023-10-24

**Soundness:** 3 good
**Presentation:** 3 good
**Contribution:** 3 good
**Rating:** 8
**Confidence:** 3

**Summary:**

This paper proposes that the existing erasure methods do not fully excise concepts from the generative models. Special prompts can be used as attack procedures to regenerate the unsafe outputs.

**Strengths:**

1. The target issues of the paper are meaningful and worth exploring. This submission gives a valuable implementation of such an idea and presents good results. Existing work rarely explores the failure cases of concept erasure methods, while it is valuable.
2. The paper is generally well-written, clearly structured, and quite easy to follow.
3. Many experiments are conducted to verify the viewpoints.

**Weaknesses:**

1. The CI attack method is straightforward. Basically, this paper uses the existing methods, such as Textual Inversion (Gal et al., 2023), for the Concept Inversion Method. The CI methods lack insightful and novel design.
2. It seems that this paper only shows a number of image cases. The experiments conducted on large-scale datasets are favored.

**Questions:**

See Weaknesses

---

> ### Author Response · Authors · 2023-11-18
> **Response to Reviewer h1Qn**
>
> We thank Reviewer h1Qn for the positive comments and insightful feedback. We are delighted that our paper's structure, clarity, and the significance of the addressed issues have been well-received. In response to your concerns, we offer the following clarifications:
>
> **The CI attack method is straightforward. Basically, this paper uses the existing methods, such as Textual Inversion (Gal et al., 2023), for the Concept Inversion Method. The CI methods lack insightful and novel design.**
>
> While we mostly utilize Textual Inversion for Concept Inversion, we do think the fact that an existing and straightforward method can circumvent current concept erasure methods emphasizes the vulnerability of post hoc erasure methods.
> We also want to note that vanilla Textual Inversion does not work for SLD and NP, and we had to invent two novel variants that can search for the appropriate soft embeddings in these methods, where Textual Inversion fails. In particular, section 4.2.3 provides the description of Concept Inversion for NP, and section C.4 in the Appendix contains the pseudocode for Concept Inversion for SLD.
>
> **It seems that this paper only shows a number of image cases. The experiments conducted on large-scale datasets are favored.**
>
> We have provided several additional qualitative results in the Appendix (Section E). We agree with you that experiments on large-scale datasets can provide a more thorough analysis of current concept erasure methods. Hence, we would like to leave experiments on larger datasets for future work, perhaps as an extended journal version of our work.

---

### Official Review · Reviewer_cgR8 · 2023-10-24

**Soundness:** 3 good
**Presentation:** 3 good
**Contribution:** 3 good
**Rating:** 6
**Confidence:** 2

**Summary:**

Text-to-image generative models, widely adopted by the general public, present significant risks. These models can be exploited to generate sexually explicit content, mirror artistic styles without authorization, or even fabricate celebrity likenesses. A variety of methods have been proposed to "erase" sensitive concepts from these text-to-image models, some of which claim to "permanently remove targeted concepts from the weights." Empirical evidence across multiple test instances and use-cases like object removal and artistic style forgetting seem to support the efficacy of these erasure methods. However, the authors demonstrate that such post hoc concept erasure techniques are flawed and can be circumvented. They argue that these techniques essentially perform a form of input filtering, rendering them vulnerable to some more sophisticated text prompts. The authors introduce the "Concept Inversion" attack technique to recover erased concepts, effectively defeating seven recently announced hoc concept erasure methods across multiple use-cases. Their study calls into question the effectiveness of existing erasure methods and introduces a strong new evaluation methodology for future concept erasure research.

**Strengths:**

1. The authors offer a comprehensive introduction to the background, maintaining clear logic throughout the paper. Through diverse and reasonable experimental settings and evaluation methods that cover four different attack scenarios, the overall experiments are highly convincing, backed not only by objective numerical data but also by subjective evaluations from volunteers.

2. The experimental section of the article is detailed. The authors clearly expound on the principles behind the latest seven post hoc concept erasure methods and introduce adaptive concept inversion methods tailored for each. Each attack scenario is deeply analyzed, with proposed attack methods that are both clear and logical.

3. The authors' concept inversion method is effective, successfully circumventing most of the advanced Post hoc concept erasure methods. Even in instances where they could not break the SLD-Max method, as depicted in Appendix Figure 9, they provide a reasonable explanation in Appendix B.5.

**Weaknesses:**

1. The authors propose their attack methods under relatively lenient conditions, as evidenced by assumptions (1) and (2) in the PRELIMINARIES section. These assumptions are closer to a white-box setting. However, for most MLaaS (Machine Learning as a Service) platforms, attackers would not have access to the internal details of the model, let alone the ability to add new tokens to its dictionary for executing Concept Inversion attacks. The TRANSFERABILITY method outlined by the authors in Appendix 4.4 also seems to be non-transferable to a black-box environment due to the inability of the attacker to modify the model's internal dictionary. Therefore, the current CI attacks appear to still have limited effectiveness in a black-box setting.

**Questions:**

See weaknesses.

---

> ### Author Response · Authors · 2023-11-18
> **Response to Reviewer cgR8**
>
> We thank Reviewer cgR8 for the supportive remarks and insightful feedback. We are particularly encouraged by your acknowledgment of the clear logic, comprehensive experimental setup, and the effectiveness of the Concept Inversion method in our work. We would like to address your concerns below:
>
> **The authors propose their attack methods under relatively lenient conditions, as evidenced by assumptions (1) and (2) in the PRELIMINARIES section. These assumptions are closer to a white-box setting. However, for most MLaaS (Machine Learning as a Service) platforms, attackers would not have access to the internal details of the model, let alone the ability to add new tokens to its dictionary for executing Concept Inversion attacks. The TRANSFERABILITY method outlined by the authors in Appendix 4.4 also seems to be non-transferable to a black-box environment due to the inability of the attacker to modify the model's internal dictionary. Therefore, the current CI attacks appear to still have limited effectiveness in a black-box setting.**
>
> We agree that our attack scenario is closer to the white-box setting. We would like to point out that we want to use Concept Inversion attacks to better understand current erasure methods through their failure cases. Our results on Concept Inversion and the transferability of the learned embeddings demonstrate that current concept erasure methods are not fully excising the targeted concept but rather performing some input filtering. We have updated the main draft to clarify this better.

---

> > ### Comment · Reviewer_cgR8 · 2023-11-21
> >
> > Thank you for your direct and clear response to my questions. After reconsidering the role and significance of this paper, I acknowledge, as the authors have stated, that this is an area that has yet to be fully explored. Especially when current defense strategies claim to fully excise the targeted concept, this paper timely reveals a previously overlooked shortcoming in prior works. More work can be built upon this perspective. Hence, I have decided to maintain my positive score.

---

### Official Review · Reviewer_YpMN · 2023-10-31

**Soundness:** 3 good
**Presentation:** 3 good
**Contribution:** 3 good
**Rating:** 8
**Confidence:** 4

**Summary:**

This paper presents a comprehensive analysis of concept erasure techniques in the context of text-to-image diffusion models. The authors investigate the effectiveness of seven existing methods, which encompass both fine-tuning-based and inference-guiding-based approaches. Through their proposed concept inversion technique, the authors demonstrate that these current methods fall short in completely removing target concepts.

**Strengths:**

1. The motivation and storyline are reasonable and novel. The organization and most of the writing are clear, ensuring easy comprehension. The inclusion of preliminary experiments on Text Inversion is appreciated, as it demonstrates that the text inversion is not able to introduce additional knowledge to the model.

2. The authors conduct extensive experiments on the existing 7 works of erasure methods. The paper presents both qualitative and quantitative results, further enhancing its robustness.

**Weaknesses:**

1. While the concept inversion attack in this paper assumes full access to the diffusion model and erasure method, which may be seen as a relatively strong assumption, the reviewer acknowledges that the current setting still represents a significant advancement towards enhancing the safety of text-to-image (T2I) models. Therefore, this limitation can be considered a minor weakness rather than a significant drawback.

2. In Section 4.4, there are several points of misunderstanding that arise, pls see questions.

**Questions:**

1. There is curiosity about the behavior of the learned concept placeholder embedding in comparison to the embedding of the original concept name. This comparison could provide valuable insights into the effectiveness of the proposed approach.

2. Typo: inconsistency in the caption of Figure 7, which does not align with the order of the figure.

3. confusion regarding Section 4.4

* How to understand ‘remapping the concept in token space’? Since the original text encoder and the embedding of the original text tokens are not updated, right?

* how the authors conducted experiments to demonstrate the editability, as the left figure of Figure 7 does not appear to clearly showcase this aspect?

---

> ### Author Response · Authors · 2023-11-18
> **Response to Reviewer YpMN**
>
> We are grateful to Reviewer YpMN for their positive and insightful assessment of our work. Your recognition of the novelty, clarity, and the comprehensive nature of our experiments is highly appreciated. We would like to address your concerns to further clarify the aspects of our research below:
>
> **There is curiosity about the behavior of the learned concept placeholder embedding in comparison to the embedding of the original concept name. This comparison could provide valuable insights into the effectiveness of the proposed approach.**
>
> We agree this is an important question. While optimizing for concept placeholder embedding which is very close to the initial concept does result in deteriorated performance, the placeholder embeddings further away from the original word behave as well within a sentence as the original word embeddings (Please see Figure 7). We leave further investigation of the properties with these embeddings for future work.
>
> **Typo: inconsistency in the caption of Figure 7, which does not align with the order of the figure.**
>
> We have updated the caption to better align with the figure.
>
> **How to understand ‘remapping the concept in token space’? Since the original text encoder and the embedding of the original text tokens are not updated, right?**
>
> You are absolutely correct that the original text encoder and the embedding of the original text tokens are not updated. What we are trying to convey is that current concept erasure methods are suppressing the generation of the targeted concept when conditioned on the input embeddings corresponding to the concept text, i.e. Van Gogh. However, there exist word embeddings that trigger the generation of the erased concept in SD 1.4 that also work on the erased model. This means that some degree of input-filtering is occurring. Moreover, since the behavior of the post-erasure model is changed due to fine-tuning the UNet or interference of the generation process, we believe that the erased concept is also mapped to some input embeddings that do not necessarily transfer to the pre-erasure model. We will add further clarifications in the revised manuscript.
>
> **how the authors conducted experiments to demonstrate the editability, as the left figure of Figure 7 does not appear to clearly showcase this aspect?**
>
> Following Gal et al. 2022, we conducted the editability experiment by first performing Concept Inversion on the erased model, and then generating images using prompts containing the special learned token (e.g. “A painting of the beach in the style of <special-token>"). We then measure the CLIP [1] similarity between the generated images and the text prompt without the special token (e.g.  “A painting of the beach”). This allows us to evaluate whether the learned embeddings can be used in various contexts.
>
> *__Reference:__*
>
> [1] Radford, Alec, et al. "Learning transferable visual models from natural language supervision." International Conference on Machine Learning, 2021.

---

> > ### Comment · Reviewer_YpMN · 2023-11-21
> > **Thanks for the rebuttal.**
> >
> > I acknowledge that I have read the rebuttal, and all my concerns are addressed.
> > I maintain my recommendation for acceptance.

---

### Official Review · Reviewer_PvZ8 · 2023-10-31

**Soundness:** 2 fair
**Presentation:** 3 good
**Contribution:** 2 fair
**Rating:** 5
**Confidence:** 4

**Summary:**

The paper reveals that seven recent concept erasure methods in text-to-image models are ineffective, as erased concepts can be regenerated with specialized prompts. It demonstrates the methods' vulnerability and the need for stronger evaluation approaches to truly sanitize models from sensitive content.

**Strengths:**

1. The paper is clear and well-structured.
2. Experiments are well-executed.
3. The authors have provided the accompanying code.

**Weaknesses:**

1. The paper evaluates black-box methods using a white-box approach. For example, Kumari et al. [1] already acknowledged their limitation in the white-box setting.

2. Clarification is needed regarding the attack scenarios presented. Specifically, it is unclear why an adversary with white-box access would seek to bypass a black-box model.

3. Assumption 5 in section 3 appears inconsistent with a white-box setting, particularly when considering Assumption 4, which grants the adversary computational resources. It raises the question of why an adversary could not alter the weights of the "erased" model under these conditions.

4. Regarding the practicality of the proposed attack method, it is important to note that widely-used public services such as Stability AI and DALL-E 3 do not currently accept text-embedding inputs to my knowledge. This may render the evaluation less critical, especially considering that existing methods effectively erase explicitly mentioned words.

[1] Kumari et al., Ablating Concepts in Text-to-Image Diffusion Models ICCV 2023

**Questions:**

Please refer weakness part.

---

> ### Author Response · Authors · 2023-11-18
> **Response to Reviewer PvZ8**
>
> We thank Reviewer PvZ8 for the constructive comments and insightful feedback. We are pleased to know that the reviewer finds our paper clear and well-structured, and our experiments rigorous. We would like to address the concerns below:
>
> **The paper evaluates black-box methods using a white-box approach. For example, Kumari et al. [1] already acknowledged their limitation in the white-box setting.**
>
> We fully acknowledge the transparency by Kumari et al. [1] in the white-box setting. Nevertheless, this limitation has not been clearly stated in the remaining concept erasure methods. Moreover, while Kumari et al. mention that their method does not prevent the re-introduction of the erased concept in the white-box setting, we are the first to point out that it is their method (and 6 other concept erasure methods covered in our work) mostly provide no protection against more sophisticated inputs.
>
> **Clarification is needed regarding the attack scenarios presented. Specifically, it is unclear why an adversary with white-box access would seek to bypass a black-box model.**
>
> Through an adversary with white-box access, our work aims to understand better the failure cases of concept erasure methods. Our experiments reveal that the seemingly erased concepts still persist within the model post-erasure. Hence, a malicious user can exploit this false sense of security and trigger the erased model to generate sensitive content.
>
> **Assumption 5 in section 3 appears inconsistent with a white-box setting, particularly when considering Assumption 4, which grants the adversary computational resources. It raises the question of why an adversary could not alter the weights of the "erased" model under these conditions.**
>
> We would like to clarify that we are not merely trying to generate erased concepts from sanitized models. From a scientific perspective, our goal in this paper is to investigate two questions:
>
>  (1) does the erased concept still persist in the model post-erasure?
>
>  (2) are current concept erasure methods performing input filtering?
>
> While fine-tuning the entire model on samples containing the erased concept can re-introduce that concept back into the model, it does not help us answer the previously mentioned questions. Hence, we want to restrict the adversary from altering the weights of the erased model.
>
> **Regarding the practicality of the proposed attack method, it is important to note that widely-used public services such as Stability AI and DALL-E 3 do not currently accept text-embedding inputs to my knowledge. This may render the evaluation less critical, especially considering that existing methods effectively erase explicitly mentioned words.**
>
> We agree with the reviewer regarding the practicality of our proposed attack method towards closed models which are available only through API access. However, the main goal of our work is to take a step toward understanding current concept erasure methods and their vulnerabilities. We hope that our work can shed light on the limitations of these methods to the research community, which can encourage future works to focus more on practical attacks and appropriate defenses. Moreover, our experiments with white-box access demonstrate that (1) current concept erasure methods are susceptible to soft-embedding attacks, and (2) there exists soft-embeddings that can trigger both the original (unerased) model and the fine-tuned (erased) model to generate the target concepts. One possible attack under the black-box assumption is exhaustively performing discrete optimization (black-box) attacks on the unerased model until we find the hard prompts that “jailbreak” the erased model. We leave further explorations of black-box attacks for future work.
>
> *__Reference:__*
>
> [1] Kumari et al., “Ablating Concepts in Text-to-Image Diffusion Models”, International Conference on Computer Vision, 2023.

---

> > ### Comment · Reviewer_PvZ8 · 2023-11-22
> >
> > Thank you for your response and for the engaging discussion on your research methodology.
> >
> > I concur with your perspective that the primary focus of your method lies in the evaluation of current concept ablation methods. This emphasis on evaluation, rather than on attack methodologies, seems to be a more accurate representation of your research goals. It's an important distinction that highlights the evaluative strength of your approach in understanding and assessing these methods.

---

> > > ### Author Response · Authors · 2023-11-22
> > >
> > > We thank the reviewer for continuing the discussion. Given that the reviewer has now a more positive outlook on our work, we would appreciate a re-evaluation of the rating.

---

### Official Review · Reviewer_iwRd · 2023-10-31

**Soundness:** 3 good
**Presentation:** 3 good
**Contribution:** 3 good
**Rating:** 6
**Confidence:** 3

**Summary:**

This paper examines seven concept erasure methods in text-to-image models, and show that the targeted concepts that are supposedly erased from the models can be retrieved using an algorithm which learns special input word embeddings. In this way, this paper shows more thoughts and efforts are needed when coming up with concept erasure methods in text-to-image models.

**Strengths:**

The paper addresses an important problem that is often not looked upon in text-to-image generation models.

Visual illustrations are presented well. It is also good that both quantitative and qualitative results are shown.

Detailed appendix is also helpful.

**Weaknesses:**

Only 1 version of Stable Diffusion (version 1.4) is presented. Is there any reason to choose this? Will the results be similar in later versions of Stable Diffusion?

Though the paper exposes that the seven recently proposed concept erasure methods can be broken, the paper does not address in detail how these can be fixed. A detailed discussion section on this will be useful and make the paper stronger.

**Questions:**

None.

---

> ### Author Response · Authors · 2023-11-18
> **Response to Reviewer iwRd**
>
> We sincerely thank Reviewer iwRd for their constructive comments and valuable feedback on our submission. We are gratified that the reviewer acknowledges the importance of the problem our paper addresses in the context of text-to-image generation models, and appreciates the clarity of our visual illustrations, the comprehensive nature of our results, and the detailed appendix provided. Below, we address the concerns raised to further clarify our work:
>
> **Only 1 version of Stable Diffusion (version 1.4) is presented. Is there any reason to choose this? Will the results be similar in later versions of Stable Diffusion?**
>
> The reason why we only consider SD 1.4 is because nearly all results of 7 concept erasure methods used SD 1.4. We have provided additional results of applying AC [1] and SLD [2] on SD 2.0 as well as performing Concept Inversion on the corresponding erased models. Please see Section F of the Appendix in the revised manuscript.
>
> **Though the paper exposes that the seven recently proposed concept erasure methods can be broken, the paper does not address in detail how these can be fixed. A detailed discussion section on this will be useful and make the paper stronger.**
>
> We have added a discussion section in the Appendix on potential solutions to make concept erasure methods more effective and robust against adversarial inputs. Please see Section A of the Appendix in the revised manuscript for the discussion.
>
> *__Reference:__*
>
> [1] Kumari et al., “Ablating Concepts in Text-to-Image Diffusion Models”, International Conference on Computer Vision, 2023.
>
> [2] Schramowski, Patrick, et al. "Safe latent diffusion: Mitigating inappropriate degeneration in diffusion models.", Conference on Computer Vision and Pattern Recognition, 2023.

---

### Author Response · Authors · 2023-11-18
**General Response and Summary of Updates to Manuscript**

We thank the reviewers for noting that we address an important and well-motivated problem (iwRd, YpMN, h1Qn), with a comprehensive as well as well-executed series of experiments (iwRd, PvZ8, YpMN, cgR8, h1Qn) that is well organized, written, and easy to follow (PvZ8, YpMN, cgR8, h1Qn).

-------

Here is the summary of updates that we've made to the manuscript:

- Added additional results on SD 2.0 in Section F of the Appendix.
- Added a discussion section of our attack scenario and potential solutions to make text-to-image models more robust in Section A of the Appendix.
- Fixed the caption of Figure 7 to align with the presented figures.
- Updated Section 4.4 to clarify the input filtering phenomenon.
- Added additional qualitative results in Section E of the Appendix.

---

### Meta-Review · Area_Chair_1A3N · 2023-12-06

**Metareview:**

This paper addresses some of the drawbacks of text-to-image generative models, specifically they can produce copyrighted or pornographic content without proper guardrails. Many guardrails rely on erasure techniques, which have vulnerabilities.  This paper shows how such erasures can be recovered and guardrails violated. The authors argue that these techniques essentially perform a form of input filtering, rendering them vulnerable to some more sophisticated text prompts. This paper introduces the "Concept Inversion" attack technique to recover erased concepts, effectively defeating seven recently announced hoc concept erasure methods across multiple use-cases. Their study calls into question the effectiveness of existing erasure methods and introduces a strong new evaluation methodology for future concept erasure research.

Strengths

The paper addresses an important problem that is often not looked upon in text-to-image generation models.
The paper is clear and well-structured.
Experiments are well-executed.
The authors have provided the accompanying code.
Both quantitative and qualitative results are shown.
Visuals and the detailed appendix are considered  helpful.

Weaknesses

Though the paper exposes that the seven recently proposed concept erasure methods can be broken, the paper does not address in detail how these can be fixed. A detailed discussion section on this will be useful and make the paper stronger.

The title therefore can be considered misleading aas at least one reviewer feels that it implies "that the primary focus of the proposed method is on attacks, even though the authors' primary emphasis appears to be on evaluation".

**Justification For Why Not Higher Score:**

The paper is not without weaknesses, PvZ8, cgR8 and YpMN all agree that the attack assumptions and the limitations of evaluation (only testing with one version of stable diffusion) reduce its importance and impact.

**Justification For Why Not Lower Score:**

The majority of the reviewers agree to accept the paper, and the authors rebuttal answered all of the concerns of at least one reviewer who gave the paper only a "6" instead of an "8."

---

### Decision · Program_Chairs · 2024-01-16

Accept (poster)